# Chondrocytes in the resting zone of the growth plate are maintained in a Wnt-inhibitory environment

**Shawn A Hallett[1], Yuki Matsushita[1], Wanida Ono[1,2], Naoko Sakagami[1], Koji Mizuhashi[1], Nicha Tokavanich[1], Mizuki Nagata[1], Annabelle Zhou[1], Takao Hirai[3], Henry M Kronenberg[4], Noriaki Ono[1,2]***

[1]University of Michigan School of Dentistry, Ann Arbor, United States; [2]University of Texas Health Science Center at Houston School of Dentistry, Houston, United States; [3]Ishikawa Prefectural Nursing University, Ishikawa, Japan; [4]Endocrine Unit, Massachusetts General Hospital and Harvard Medical School, Boston, United States

**Abstract** Chondrocytes in the resting zone of the postnatal growth plate are characterized by slow cell cycle progression, and encompass a population of parathyroid hormone-related protein (PTHrP)-expressing skeletal stem cells that contribute to the formation of columnar chondrocytes. However, how these chondrocytes are maintained in the resting zone remains undefined. We undertook a genetic pulse-chase approach to isolate slow cycling, label-retaining chondrocytes (LRCs) using a chondrocyte-specific doxycycline-controllable Tet-Off system regulating expression of histone 2B-linked GFP. Comparative RNA-seq analysis identified significant enrichment of inhibitors and activators for Wnt signaling in LRCs and non-LRCs, respectively. Activation of Wnt/β-catenin signaling in PTHrP[+] resting chondrocytes using *Pthlh-creER* and *Apc*-floxed allele impaired their ability to form columnar chondrocytes. Therefore, slow-cycling chondrocytes are maintained in a Wnt-inhibitory environment within the resting zone, unraveling a novel mechanism regulating maintenance and differentiation of PTHrP[+] skeletal stem cells of the postnatal growth plate.

*For correspondence:
noriaki.ono@uth.tmc.edu

**Competing interests:** The authors declare that no competing interests exist.

## Introduction

The epiphyseal growth plate, a disk of cartilaginous tissues with characteristic columns of chondrocytes formed between the primary and secondary ossification centers, is an innovation of amniotes (reptiles, birds and mammals) that facilitates explosive endochondral bone growth (*Wuelling and Vortkamp, 2019*). The postnatal growth plate is composed of three morphologically distinct layers of resting, proliferating and hypertrophic zones, in which chondrocytes continue to proliferate well into adulthood, especially in mice, therefore functioning as the engine for endochondral bone growth (*Hallett et al., 2019*; *Kronenberg, 2003*). By adulthood, a large majority of hypertrophic chondrocytes undergo apoptosis or transdifferentiate into osteoblasts, marking the completion of the longitudinal growth phase and skeletal maturation (*Zhou et al., 2014*; *Park et al., 2015*; *Hallett et al., 2021*; *Roach et al., 1995*; *Yang et al., 2014a*; *Yang et al., 2014b*).

Of the three layers, the resting zone has two important functions in maintaining the growth plate. First, early studies postulated that resting chondrocytes feed their daughter cells into the adjacent proliferating zone and contribute to longitudinal growth of postnatal endochondral bones (*Hunziker, 1994*). More recently, the resting zone has been established as a niche for skeletal stem cells, initially demonstrated by surgical auto-transplantation experiments in rabbits (*Abad et al., 2002*), and subsequently by lineage-tracing experiments in mice (*Mizuhashi et al., 2018*; *Newton et al., 2019*). Second, chondrocytes in the resting zone express parathyroid hormone-related protein (PTHrP) that maintains proliferation of chondrocytes in a cell non-autonomous manner and delays

their hypertrophic differentiation, thus sustaining longitudinal growth (*Schipani et al., 1997*). Once resting chondrocytes exit their quiescent state, they undergo a morphological transformation into flat, columnar proliferating chondrocytes. The proliferating zone is concertedly maintained by PTHrP released from the resting zone and Indian hedgehog (Ihh) synthesized by pre-hypertrophic chondrocytes; the proliferating zone in turn provides instructive cues to regulate cell fates of PTHrP$^+$ chondrocytes (*Mizuhashi et al., 2018*). Therefore, the resting zone functions as a critical constituent of the tight feedback system (the PTHrP–Ihh feedback loop) that maintains growth plate structures and longitudinal bone growth.

The mechanisms regulating self-renewal and differentiation capabilities of resting zone chondrocytes remain largely unknown. G protein stimulatory subunit-$\alpha$ (G$_s\alpha$), G$_q$/G$_{11}\alpha$ G proteins, which are coupled with the PTH/PTHrP receptor (PPR), are both required for maintaining quiescent stem-like chondrocytes (*Chagin et al., 2014*). Pan-chondrocyte ablation of G$_s\alpha$ (*Col2a1-creER; Gnas$^{f/f}$*) causes premature differentiation of stem-like resting chondrocytes into the proliferative pool, resulting in accelerated endochondral bone growth. Further, combined inactivation of G$_q$/G$_{11}\alpha$ through a mutant PPR (*Guo et al., 2002*) and G$_s\alpha$ (*Col2a1-creER, Gnas$^{f/f}$*; PPR$^{D/D}$ [Pth1r$^{tm4Hmk}$]) causes a more severe phenotype associated with growth plate fusion. Therefore, PPR-mediated G$_s\alpha$ and G$_q$/G$_{11}\alpha$ synergistically maintain quiescence of resting chondrocytes and their differentiation into columnar chondrocytes (*Chagin et al., 2014*); however, whether this regulation occurs cell-autonomously in resting chondrocytes has not been determined.

Non-canonical Wnt/planar cell polarity (PCP) signaling plays a key role in facilitating asymmetric divisions of resting chondrocytes. Oriented division, rearrangement and intercalation of chondrocyte clones in the resting zone, and their subsequent asymmetric divisions into their daughter cells aligned with the axis of growth are the hallmark characteristic of growth plate chondrocytes (*Li et al., 2017a*). Non-canonical Wnt/PCP signaling is activated when the chondrocytes exit the resting zone and start forming columns, guiding the oriented cell division of resting chondrocytes into proliferating cells and their further expansion. In fact, misregulation of non-canonical Wnt/PCP signaling via dominant-negative forms of Frizzled receptors results in severe shortening of the growth plate (*Hartmann and Tabin, 2000*). Oriented cell division is sensitive to both high and low PCP activity mediated in part by Fzd7 (*Li and Dudley, 2009*; *Li et al., 2017b*). In addition, Wnt5a signals to establish PCP in chondrocytes along the proximal-distal axis through regulation of Vangl2 (*Gao et al., 2011*; *Randall et al., 2012*). However, how resting chondrocytes are regulated by non-canonical Wnt/PCP signaling members, such as Rspo3 (*Ohkawara et al., 2011*) in addition to Dkk2 (*Niehrs, 2006*) and Fzd receptors, are unknown.

Given the limited amount of mechanistic knowledge regarding maintenance and differentiation of resting chondrocytes, we set out to undertake an unbiased approach to better define the molecular mechanisms regulating maintenance and differentiation of chondrocytes in the resting zone ('slow-cycling chondrocytes'). To achieve this goal, we developed a chondrocyte-specific genetic label-retention strategy to isolate slow-cycling chondrocytes from the postnatal growth plate. Our comparative transcriptomic analysis revealed unique molecular signatures defining the characteristics of slow-cycling chondrocytes, with particular enrichment for inhibitors of Wnt signaling pathways. Subsequent functional validation based on a cell-lineage analysis identified that, when Wnt/$\beta$-catenin signaling was activated, PTHrP$^+$ resting chondrocytes were decreased in number during initial formation and established columnar chondrocytes less effectively in the subsequent stages. These data lead to a new concept that PTHrP$^+$ skeletal stem cells may be maintained in a Wnt inhibitory environment within the resting zone niche of the postnatal growth plate.

## Results

### A genetic label-retention strategy to identify slow-cycling chondrocytes

Chondrocytes in the resting zone of the postnatal growth plate ('resting' or 'reserve' chondrocytes) are characterized by their slow cell cycle progression that is much slower than that of chondrocytes in the proliferating zone. As a result, these slow-cycling chondrocytes retain nuclear labels much longer than their more rapidly dividing progeny in the proliferating zone, which are therefore termed as label-retaining chondrocytes (LRCs) (*Walker and Kember, 1972*). First, we undertook a genetic approach to fluorescently isolate LRCs from the growth plate based on a chondrocyte-specific pulse-

chase protocol. To this end, we first generated transgenic mice expressing a tetracycline-controlled transactivator under a *Col2a1* promoter (hereafter, Col2a1-tTA), and combined this line with a *Col1a1* locus harboring a Tet-responsive element (TRE)-histone 2B-bound EGFP (H2B-EGFP) cassette

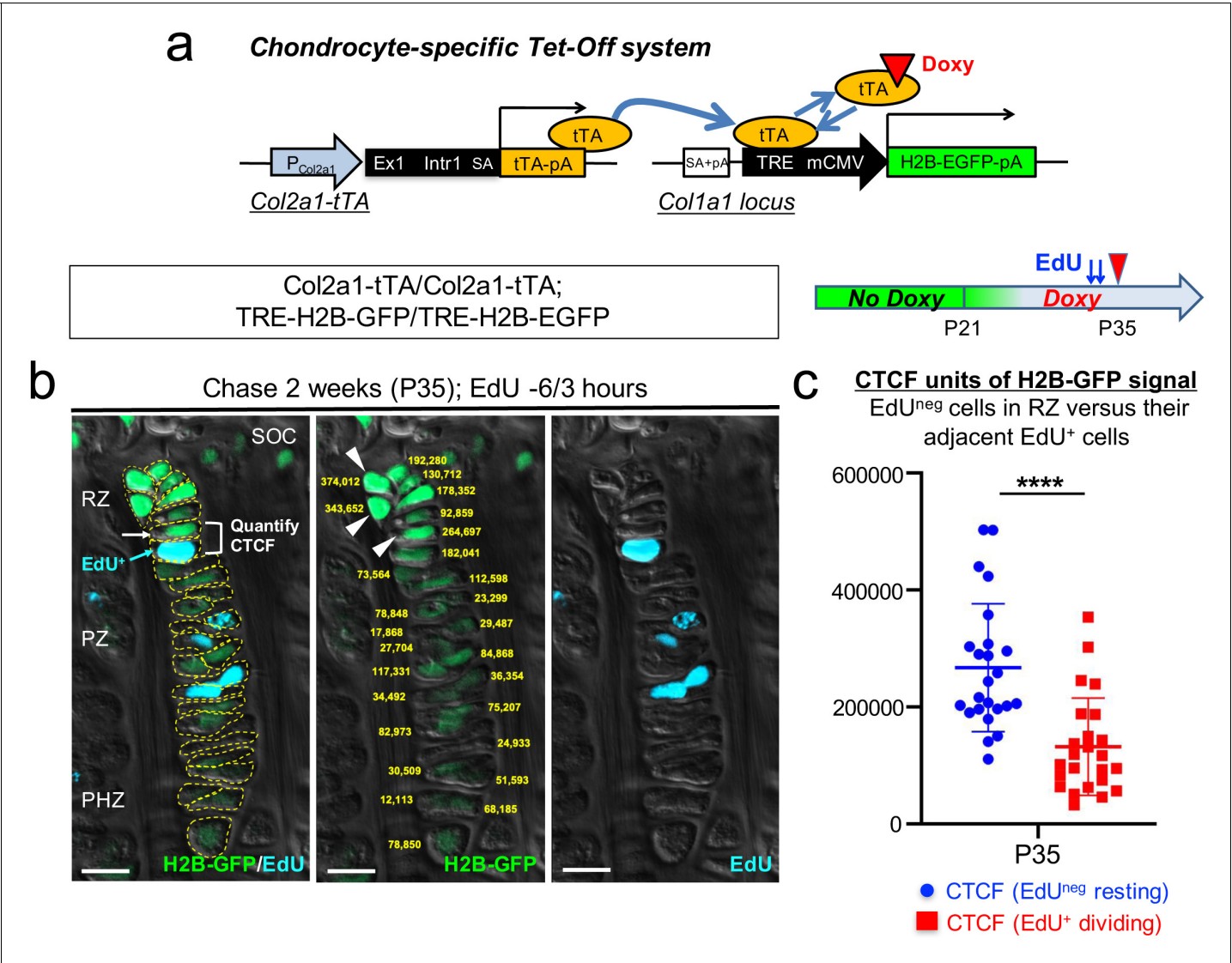

**Figure 1.** A genetic label-retention strategy to identify slow-cycling chondrocytes. (a) Chondrocyte-specific Tet-Off system by *Col2a1-tTA* and *TRE-H2B-EGFP* transgenes. During development, *Col2a1*+ cells accumulate H2B-EGFP in the nucleus. Binding of tetracycline controlled transactivator (tTA) to Tet-responsive element (TRE) is prevented in the presence of doxycycline. As a result of this chase, slow-cycling cells retain a high level of H2B-EGFP, whereas proliferating cells dilute H2B-EGFP signal as they continue to divide. (b) Proximal tibial growth plates of Col2a1-tTA/Col2a1-tTA; TRE-H2B-EGFP/TRE-H2B-EGFP double homozygous mice after 2 weeks of chase at P35, with EdU administration shortly before analysis (6 and 3 hr prior to sacrifice). Shown is a representative 27 chondrocyte-long column highlighting gradient of H2B-GFP signal across different layers. (Left): Merged image of H2B-GFP, EdU-Alexa647, and DIC with tracing of individual chondrocytes. (Middle): Quantification of corrected total cell fluorescence (CTCF) values of H2B-GFP signal in chain. (Right): EdU-Alexa647 signal. SOC: secondary ossification center, RZ: resting zone, PZ: proliferating zone, PHZ: pre-hypertrophic zone. Arrows and a bracket: a pair of an H2B-GFP$^{high}$ cell in RZ (white) and its adjacent EdU+ cell (blue) in PZ. Arrowheads: the top three brightest cells in the column in RZ. Dotted lines: individual chondrocytes in column. Yellow: CTCF values. Green: H2B-GFP, blue: EdU, gray: DIC. Scale bars: 100 µm. (c) Quantification of CTCF values of H2B-GFP signal in a pair of a H2B-GFP$^{high}$ cell in RZ and its adjacent EdU+ dividing cell. Twenty-four cell pairs from *n*=four mice. ****$p<0.0001$. Mann-Whitney's *U*-test. Data are represented as mean ± s.d.

The online version of this article includes the following source data and figure supplement(s) for figure 1:

**Source data 1.** Quantification of corrected total cell fluorescence (CTCF) values of H2B-GFP signal in chain.
**Source data 2.** Quantification of CTCF values of H2B-GFP signal in a pair of a H2B-GFP$^{high}$cell in RZ and its adjacent EdU+dividing cell.
**Figure supplement 1.** A genetic label-retention strategy to identify slow-cycling chondrocytes.

(hereafter, TRE-H2B-EGFP) (*Figure 1a*, *Figure 1—figure supplement 1a*). In this Tet-Off system, tTA binds to TRE in the absence of doxycycline and activates H2B-EGFP transcription (pulse), whereas tTA dissociates from TRE in the presence of doxycycline, shutting off H2B-EGFP transcription (chase) (*Figure 1a*, *Figure 1—figure supplement 1a*). In Col2a1-tTA; TRE-H2B-EGFP mice, *Col2a1*⁺ chondrocytes accumulate H2B-EGFP in the nucleus without doxycycline, and upon initiation of doxycycline feeding, de novo transcription of *H2B (human H2B [HIST1H2BJ])-EGFP* mRNA becomes suppressed. After a long chase period, H2B-EGFP is preferentially diluted in highly proliferating cells and their progeny, whereas slow-cycling cells retain a high level of the label, marking them as LRCs.

In order to evaluate the labeling efficiency of the system, we first analyzed double heterozygous Col2a1-tTA/+; TRE-H2B-EGFP/+ mice at postnatal day (P) seven and P21 without doxycycline. While most of chondrocytes in the growth plate were marked by a high level of H2B-EGFP at P7 (*Figure 1—figure supplement 1b*), fewer than half of columnar chondrocytes in the growth plate were marked by H2B-EGFP at P21 (*Figure 1—figure supplement 1c*), demonstrating the inefficiency of the Tet system in postnatal growth plate chondrocytes. To circumvent this problem, we further generated double homozygous Col2a1-tTA/Col2a1-tTA; TRE-H2B-EGFP/TRE-H2B-EGFP mice and analyzed these mice at P21 without doxycycline. A great fraction of columnar chondrocytes was marked by H2B-EGFP (*Figure 1—figure supplement 1d*), indicating that the labeling efficiency can be improved in a transgene dosage-dependent manner in this system.

Subsequently, we tested the effectiveness of this chondrocyte-specific Tet-Off system by pulse-chase experiments. We fed double heterozygous Col2a1-tTA/+; TRE-H2B-EGFP/+ mice with doxycycline for 5 weeks starting from P21 to shut off de novo H2B-EGFP expression. We started the chase at P21 because the secondary ossification center was fully developed within the epiphysis by this time. After the chase, the H2B-EGFP signal was largely abrogated in the growth plate region, with only a small fraction of cells in the resting zone near the top of the growth plate retaining H2B-EGFP (*Figure 1—figure supplement 1e*, arrowheads). However, we also noticed that a low level of H2B-EGFP signal persisted in adjacent osteoblasts and osteocytes in the epiphysis even after the chase (*Figure 1—figure supplement 1e*, arrows), making it difficult to distinguish LRCs from these cells. Analysis of TRE-H2B-EGFP/+ mice without a Col2a1-tTA transgene at P28 revealed that osteoblasts and osteocytes expressed a low level of H2B-EGFP (*Figure 1—figure supplement 1f*, arrows). These findings indicate that LRCs can be identified within the top of the growth plate by a chondrocyte-specific Tet-Off system regulating H2B-EGFP expression, although these cells cannot be easily distinguished from adjacent osteoblasts and osteocytes solely based on fluorescent intensity in histological sections.

Next, to more rigorously define LRCs and non-LRCs in the postnatal growth plate, we quantified the intensity of H2B-EGFP signal through the entire column of chondrocytes using the corrected total cell fluorescence (CTCF) after two weeks of chase at P35 in double homozygous Col2a1-tTA/Col2a1-tTA; TRE-H2B-GFP/TRE-H2B-EGFP mice. We labeled actively proliferating chondrocytes by pulsing these mice with a thymidine analogue EdU twice at 6 and 3 hr prior to sacrifice. By this approach, we expect to visualize an EdU-positive cell in the proliferating zone that is descended from an EdU-negative cell in the resting zone through one cell division (*Figure 1b*, left panel, arrowheads and bracket). The quantification of CTCF values through the entire column revealed that the cells in the proliferating zone (PZ) had weaker H2B-EGFP signal than those of the resting zone (RZ), and the cells in the pre-hypertrophic (PHZ) and hypertrophic zones (HZ) had yet weaker H2B-EGFP signal (*Figure 1b*, center panel, yellow numbers, *Figure 1b*; *Figure 1—source data 1*). The side-by-side comparison of CTCF values between EdU⁺ cells in the proliferating zone and their preceding EdU-negative cells in the resting zone further revealed that H2B-EGFP CTCF values decreased significantly upon one cell division (*Figure 1c*, *Figure 1—source data 2*, P35: H2B-EGFP CTCF, EdU⁻ cells in RZ = 267,184.6±109,457.8; EdU⁺ cells in PZ = 131,892.0±83,344.8, *n*=four mice, 24 cell-pairs [*p*<0.0001]). The average fold change between EdU-negative and EdU-positive cells was 2.03, suggesting that H2B-EGFP intensity decreases by a factor of two following one cell division (*Figure 1b, c*), as expected from the fact that histone-bound GFP is distributed equally between the two daughter cells upon cell division. Further, the cells with the top 10% brightness were localized to RZ at the top of the growth plate; while there were 27 cells in this given column of chondrocytes, the three brightest cells (374,012; 343,652; 264,697) were localized to RZ (*Figure 1b*, center panel,

arrowheads). Thus, based on these quantitative histological data, we designated the cells with the top 10% brightness as LRCs.

## Col2-Q system: a double-color quadruple transgenic strategy to identify LRCs in the growth plate

To circumvent the technical issues associated with the leakiness of H2B-EGFP in non-chondrocytes, we further included a *Col2a1-creER* transgene that activates an *R26R-tdTomato* reporter in a tamoxifen-dependent manner, as a means to specifically mark growth plate chondrocytes (*Chen et al., 2007*; *Nagao et al., 2016*). We generated quadruple homozygous transgenic mice – 'Col2-Q' mice: *Col2a1-tTA; TRE-H2B-EGFP; Col2a1-creER; R26R-tdTomato* (*Figure 2a*) and treated these mice with tamoxifen (4 mg) twice shortly before analysis (3 and 2 days before analysis, 'short protocol') to obtain Col2a1-creER-tdTomato$^+$ cells (hereafter, Col2$^{CE}$-tdT$^+$ cells). After the pulse-chase protocol with doxycycline, LRCs are expected to be identified as cells with green nuclei with the highest brightness and red cytoplasm, which are localized in the resting zone of the growth plate (*Figure 2b,c*).

First, we analyzed Col2-Q mice at P21 without doxycycline ('No Chase'). A great majority of cells in the epiphysis, including those in the growth plate and the secondary ossification center, but not as many on the articular surface, were H2B-EGFP$^{high}$ (*Figure 2d*, cells with green nuclei). This short protocol of tamoxifen injection marked a great number of chondrocytes in the growth plate, but a much fewer number of cells in the articular cartilage (*Figure 2d*), indicating that this double-color strategy can effectively identify H2B-EGFP$^{high}$growth plate chondrocytes at this stage. Second, Col2-Q mice were fed with doxycycline from P21 to shut off new H2B-EGFP synthesis for 4 weeks (chase) and were then treated with the short protocol of tamoxifen injection After the chase, LRCs were identified at a specific location near the top of the growth plate in the resting zone, retaining a higher level of H2B-EGFP signal (*Figure 2e*, left panel). In addition, most of these H2B-EGFP$^{high}$ cells in the growth plate were simultaneously marked as Col2$^{CE}$-tdT$^+$ (*Figure 2e*, right panel, arrowhead). While chondrocytes with the brightest H2B-GFP signal were localized to the resting zone, their descendants showed increasingly diluted H2B-EGFP signals as they progressed toward the proliferating and pre-hypertrophic zones (*Figure 1b*, *Figure 2e*, right panel). Therefore, our Col2-Q quadruple transgenic strategy can effectively mark LRCs primarily in the resting zone of the postnatal growth plate.

## A protocol to preferentially isolate growth plate chondrocytes from postnatal epiphyses

We next established a protocol to harvest chondrocytes from the postnatal growth plate. We manually removed epiphyses from four long bones (bilateral distal femurs and proximal tibias [*Figure 3a*, top panel, shown is a dissected epiphysis from a tibia]). With this protocol, the growth plate was sheared at the hypertrophic layer with the remainder attached to the epiphysis. First, we quantified Col2a1$^{CE}$-tdT$^+$ cells at P49 with tamoxifen injection at 2 and 3 days prior to dissection. Importantly, we observed a significantly fewer number of Col2a1$^{CE}$-tdT$^+$ chondrocytes in the articular surface compared to those in the growth plate at P49 (*Figure 3a,b*, ; *Figure 3—source data 1*; GP=1,315.0±171.2 cells; AC=79.1±50.3 cells, *n*=nine mice [p<0.0001]), consistent with the previous finding that *Col2a1-creER* preferentially marks growth plate chondrocytes in adulthood (*Nagao et al., 2016*). Therefore, our short-chase tamoxifen protocol with *Col2a1-creER* enables preferential labeling of growth plate chondrocytes in adulthood.

Second, we digested dissected epiphyses serially with collagenase to release these cells into single-cell suspension. Five rounds of digestion completely liberated cells from the growth plate, while cells on the articular surface were largely undigested (*Figure 3a*). We further sought to determine whether collagenase digestion enables the preferential isolation of growth plate chondrocytes. To this end, we quantified Col2a1$^{CE}$-tdT$^+$ cells in the growth plate and articular surface before and after collagenase digestion at P49 (*Figure 3b*). We found a significant reduction in Col2a1$^{CE}$-tdT$^+$ chondrocytes in the growth plate after collagenase digestion (Before: GP=1,315.0±171.2, After: GP=49.8±45.2, *n*=9 [before], *n*=4 [after] mice [*p*=0.003]). In contrast, there was no change in the number of Col2a1$^{CE}$-tdT$^+$ chondrocytes in the articular cartilage following collagenase digestion (Before: AC=79.1±50.3, After: AC=85.8±45.2, *n*=9 [before], *n*=four mice [after] [*p*=0.939]). Based on

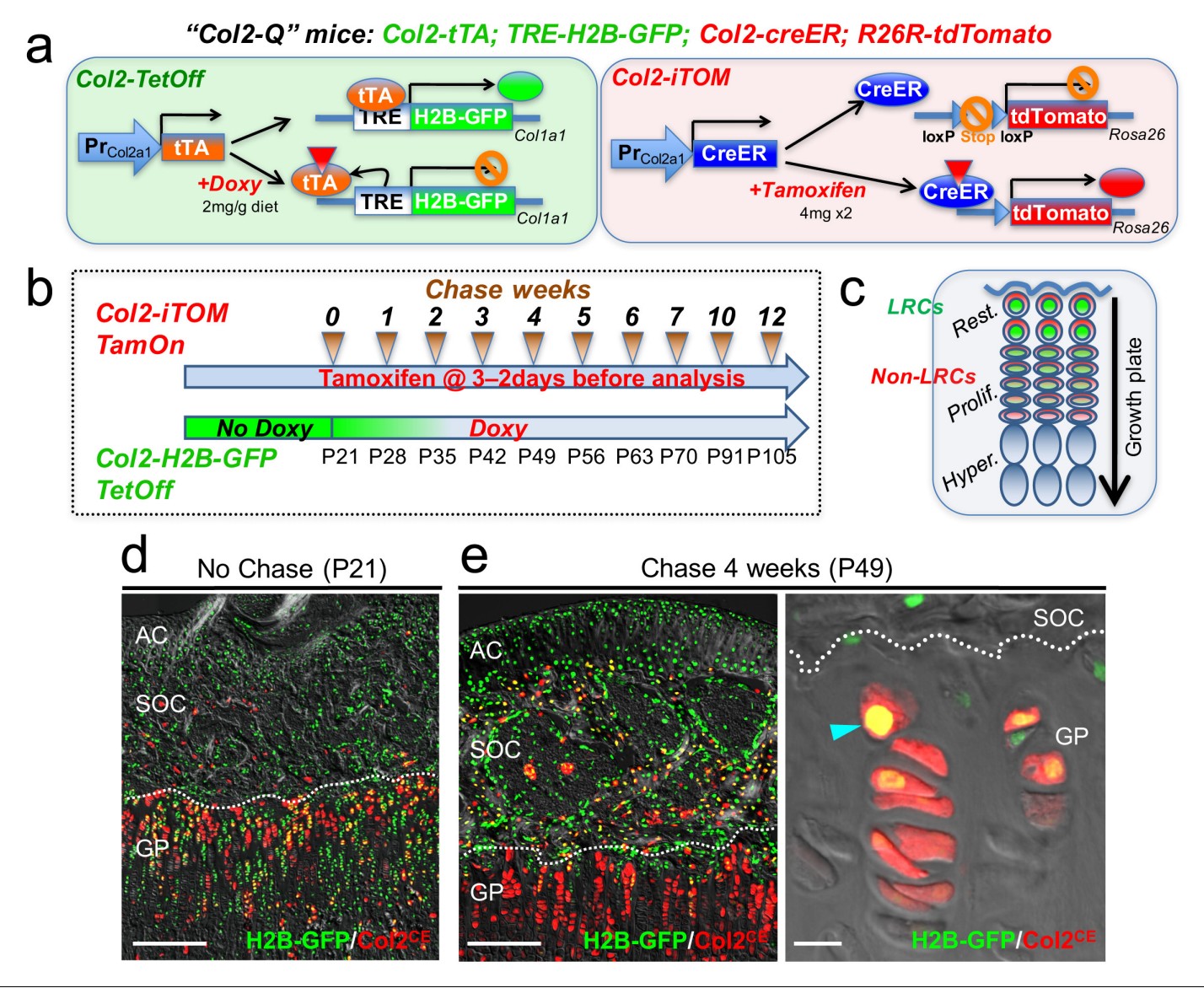

**Figure 2.** Col2-Q system: a double-color genetic label-retaining strategy to identify and isolate slow-cycling chondrocytes of the growth plate. (a) 'Col2-Q' quadruple transgenic system composed of two chondrocyte-specific bigenic Col2-Tet-Off (*Col2a1-tTA; TRE-H2B-EGFP*) and Col2-iTOM (*Col2a1-creER; R26R-tdTomato*) systems. H2B-EGFP expression can be shut off by doxycycline diet (2 mg/g diet), while tdTomato expression can be induced by two doses of tamoxifen (4 mg) administered shortly prior to analysis (3 and 2 days before). (b) Experimental design to identify label-retaining chondrocytes (LRCs) in the growth plate. Col2-Q mice are fed with doxycycline (Doxy) starting from postnatal day (P) 21 (Chase). The mice are analyzed after the indicated number of weeks; at each time point, two doses of tamoxifen are administered shortly before analysis to induce tdTomato expression. (c) Diagram for predicted outcomes. LRCs are expected to retain green nuclei with red cytoplasm and located at the resting zone. Non-LRCs have increasingly dilute levels of H2B-GFP as they become proliferative and further differentiated. Rest.: resting zone, Prolif.: proliferating zone, Hyper.: hypertrophic zone. (d,e) Col2-Q distal femur growth plates with tamoxifen injection shortly before analysis. (d): No chase, without Doxy at P21. (e): After 4 weeks of chase, on Doxy for 4 weeks at P49, right panel: high-power confocal image. Arrowhead: label-retaining chondrocytes. AC: articular cartilage, SOC: secondary ossification center, GP: growth plate. Dotted line: border between growth plate and secondary ossification center. Blue: DAPI, gray: DIC. Scale bars: 500 μm, 20 μm (fluorescent scope in **e**).

this data, we further enumerated the percentage of growth plate chondrocytes among total Col2a1$^{CE}$-tdT$^+$ cells dissociated from the dissected epiphyses. In fact, growth plate chondrocytes account for essentially all of Col2a1$^{CE}$-tdT$^+$ cells (99.1 ± 1.4% of Col2a1$^{CE}$-tdT$^+$ cells, n=four mice). Therefore, these data demonstrate that our microdissection and enzymatic dissociation approach allows us to selectively harvest Col2a1$^{CE}$-tdT$^+$ chondrocytes from the postnatal growth plate.

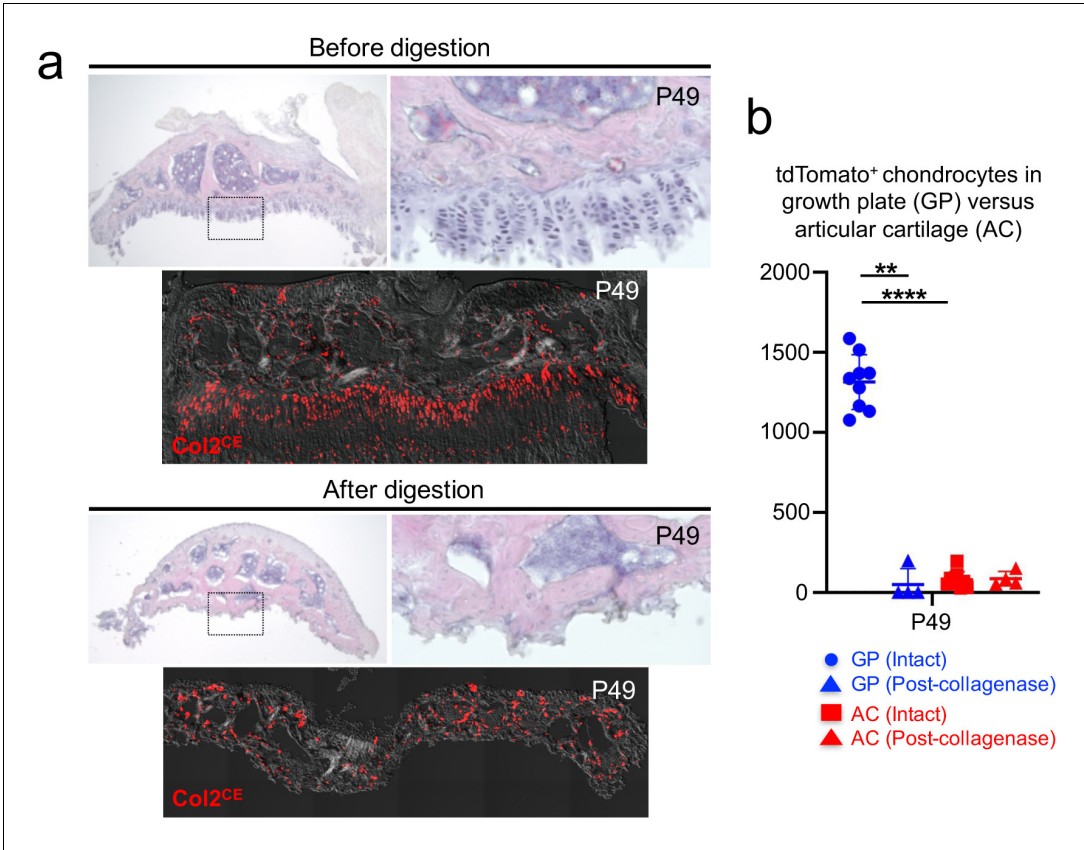

**Figure 3.** A protocol to preferentially isolate growth plate chondrocytes from postnatal epiphyses. (**a**) Representative epiphyses of *Col2a1-creER; R26R-tdTomato* proximal tibia at P49, before (upper) and after (lower) serial collagenase digestions. (Top panels): H and E staining, (bottom panels): tdTomato epifluorescence. Red: tdTomato, gray: DIC. (**b**) Quantification of tdTomato$^+$ chondrocytes in growth plate (GP) and articular cartilage (AC) before and after collagenase digestion. Blue: GP, red: AC. *n*=nine mice (Intact), *n*=four mice (Post-collagenase). **p<0.01, ****p<0.0001. Mann-Whitney's *U*-test. Data are represented as mean ± s.d. The online version of this article includes the following source data for figure 3:

**Source data 1.** Quantification of tdTomato$^+$chondrocytes in growth plate (GP) and articular cartilage (AC) before and after collagenase digestion.

## A flow cytometry-based identification and isolation of LRCs from Col2-Q mice

Subsequently, we used a flow cytometry-based approach to analyze single cells dissociated from the Col2-Q postnatal growth plate at sequential time points before and after the chase, particularly in a CD45-negative non-hematopoietic fraction. Col2a1$^{CE}$-tdT$^+$ cells were clearly distinguishable from unlabeled cells at all time points investigated (*Figure 4a*). Before the chase started at P21 (therefore without doxycycline feeding), 86.4 ± 5.0% of Col2a1$^{CE}$-tdT$^+$ cells retained >10$^4$ units of H2B-EGFP (*Figure 4a*, leftmost panel). The fraction of a label-retaining population (GFP$^{high}$, retaining >10$^4$ unit of H2B-EGFP signal) within a Col2$^{CE}$-tdT$^+$ population gradually decreased as the chase period extended (*Figure 4a,b*). These plots fit into a non-linear decay curve (Y0: 86.5 ± 1.3%; Plateau: 2.6 ± 0.9%; T$^{1/2}$=0.99~1.18 week) (*Figure 4c*, *Figure 4—source data 1*). Virtually no GFP$^+$ cells were observed in the absence of a Col2a1-tTA transgene (*Figure 4b*, magenta line), while levels of GFP$^+$ cells were maintained from five to ten weeks of chase (*Figure 4b*, orange and teal lines). Therefore, these findings demonstrate that LRCs can be effectively identified and isolated from postnatal Col2-Q growth plates by combined microdissection, enzymatic digestion and flow cytometry-based approaches.

We next set out to define whether LRCs isolated in single-cell suspension are characterized by slow-cycling nature therefore less mitotic activities. To this end, we quantitatively evaluated

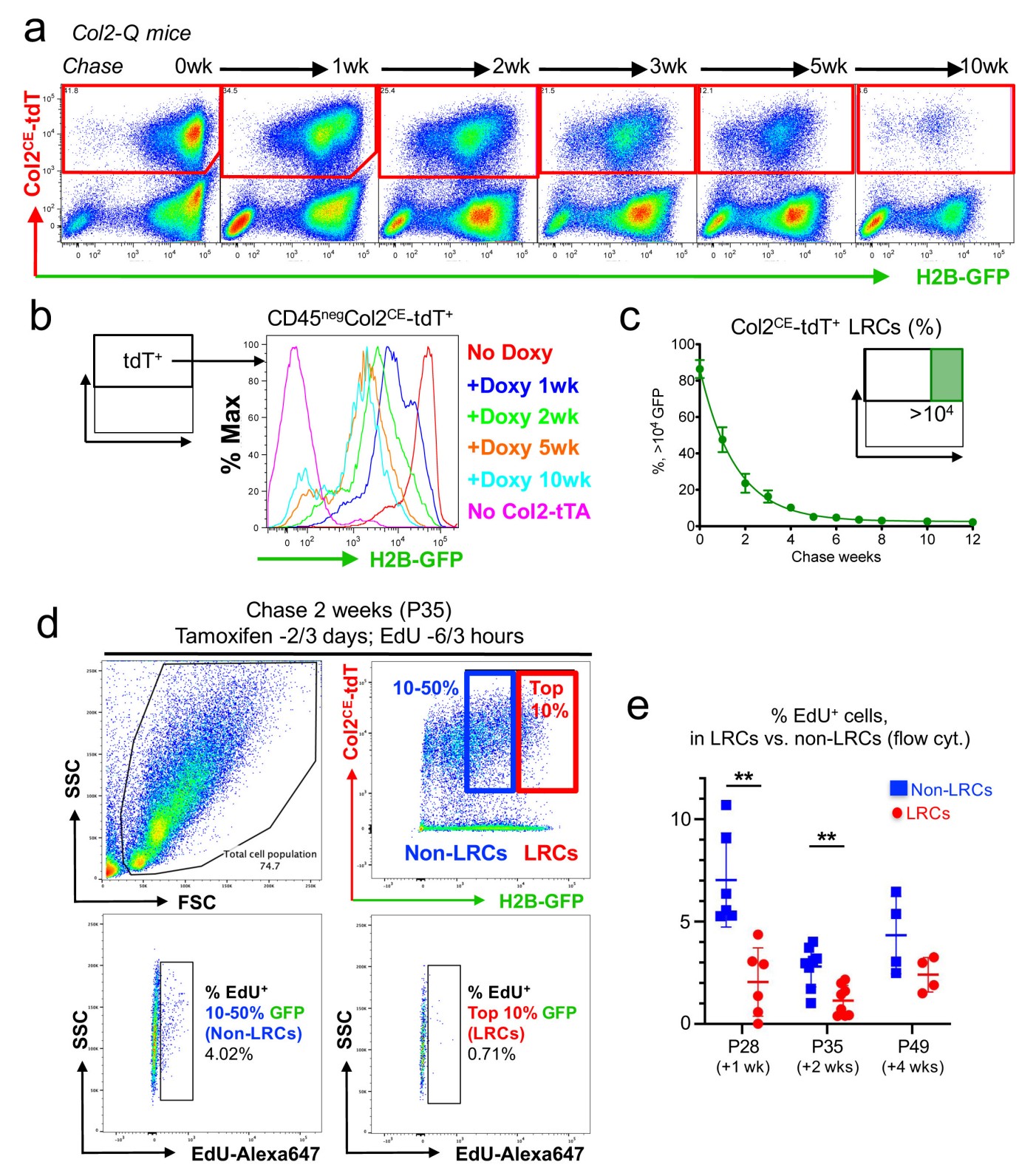

**Figure 4.** A flow cytometry-based identification and isolation of LRCs from Col2-Q mice. (**a–c**) Flow cytometry analysis of dissociated Col2-Q growth plate cells. (**a**): Pseudo-color plots of CD45$^{neg}$ cells at the indicated number of weeks in chase. Red gates: Col2a1-creER/tdTomato$^+$ (Col2$^{CE}$-tdT$^+$) cells. (**b**): Histogram of CD45$^{neg}$Col2$^{CE}$-tdT$^+$ cells showing the distribution of H2B-EGFP$^+$ cells as the percentage of the maximum count. Red line: P21 (No Doxy), blue line: P28 (+Doxy 1wk), green line: P35 (+Doxy 2wk), orange line: P56 (+Doxy 5wk), light blue line: P91 (+Doxy 10wk), pink line: No Col2-tTA

*Figure 4 continued on next page*

*Figure 4 continued*

control at P21. (c): Percentage of $>10^4$ H2B-EGFP$^+$ LRCs among total Col2$^{CE}$-tdT$^+$ cells. x axis: weeks in chase, y axis: % of cells $> 10^4$ unit of GFP. n=nine mice (0 week, 1 week), n=seven mice (2 weeks, 5 weeks), n=six mice (3 weeks, 4 weeks), n=five mice (6 weeks) and n=three mice (7 weeks, 8 weeks, 10 weeks, 12 weeks). Data are presented as mean ± s.d. (d) Flow cytometry analysis of cell proliferation in LRCs and Non-LRCs of Col2-Q growth plates at P35, with EdU administration shortly before analysis (6 and 3 hr prior to sacrifice). LRCs are defined as Col2$^{CE}$-tdT$^+$ cells with the top 10 percentile H2B-GFP brightness (red box), while Non-LRCs are defined as Col2$^{CE}$-tdT$^+$ cells with 10–50 percentile H2B-GFP brightness. Top left panel: forward/side scatter plot, top right panel: CD45-eFlour450$^{neg}$ fraction. Bottom panels: EdU-Alexa647 (x axis) signal of LRCs (right) and Non-LRCs (left). (e) Quantification of % EdU-Alexa647$^+$ cells among LRCs and non-LRCs, harvested from Col2-Q growth plates at P28 (+1 week; n=six mice), P35 (+2 weeks; n=eight mice) and P49 (+4 weeks; n=four mice). **p<0.01. Mann-Whitney's U-test. Data are represented as mean ± s.d.

The online version of this article includes the following source data and figure supplement(s) for figure 4:

**Source data 1.** Percentage of $>10^4$H2B-EGFP$^+$LRCs among total Col2$^{CE}$-tdT$^+$cells.

**Source data 2.** Quantification of % EdU-Alexa647$^+$cells among LRCs and non-LRCs.

**Figure supplement 1.** A flow cytometry-based approach to define cell proliferation of LRCs and Non-LRCs.

Col2a1$^{CE}$-tdT$^+$ cells for EdU incorporation by flow cytometry. To this end, Col2-Q mice were fed with doxycycline for 1, 2, and 4 weeks and treated with tamoxifen twice at 3 and 2 days before analysis. These mice were pulsed with two doses of EdU shortly before analysis, at 6 and 3 hr prior to sacrifice. Actively proliferating cells incorporating EdU were fluorescently marked by Click-iT Plus Alexa 647 Flow Cytometry kit. We quantified the percentage of EdU-Alexa647$^+$ cells among LRCs (cells with top 10% H2B-EGFP brightness) and non-LRCs (cells with 10–50% H2B-EGFP brightness). Non-LRCs were significantly enriched for EdU$^+$ cells, both at 1 week and 2 weeks of chase (EdU$^+$ cells, 1 week: 2.04 ± 1.67% of LRCs; 7.04 ± 2.30% of non-LRCs, n=six mice [p=0.002]; 2 weeks: 1.13 ± 0.74% of LRCs; 2.80 ± 1.00% of non-LRCs, n=eight mice [p=0.047]) (*Figure 4d,e* left and middle, *Figure 4—figure supplement 1a*, ; *Figure 4—source data 2e*). We also found a trend that non-LRCs were enriched for EdU$^+$ cells at 4 weeks of chase, although not statistically significant (EdU$^+$ cells, 2.40 ± 0.85% of LRCs; 4.34 ± 1.89% of non-LRCs, n=four mice [p=0.200] (*Figure 4e*, right, *Figure 4—figure supplement 1b*, *Figure 4—source data 2e*)). Therefore, LRCs are relatively resistant to EdU incorporation, denoting their relatively quiescent status.

## A comparative RNA-seq analysis reveals a unique molecular signature of LRCs

Subsequently, we isolated slow-cycling chondrocytes using fluorescence-activated cell sorting (FACS) at a 4-week-chase time point, when the GFP$^{high}$ label-retaining fraction ($>10^4$ unit) was sufficiently enriched (*Figure 4c*). In this experiment, LRCs were defined as GFP$^{high}$ cells retaining H2B-EGFP signal at the top 10% brightness ($x > 10^4$ unit), whereas non-LRCs were defined as other GFP$^{mid-low}$ cells ($10^3 < x < 10^4$ unit). Cells were collected from multiple littermate mice for each of three independent experiments. To assess RNA quality, we conducted an RNA Integrity Number (RIN) assay (*Schroeder et al., 2006*) from total RNAs isolated from LRCs and non-LRCs. Cellular RNA levels from each population had sufficient quality for downstream application (RIN>8.0), which were further subjected to amplification and deep sequencing. An unsupervised clustering analysis demonstrated that LRCs and non-LRCs biological triplicate samples each clustered together (*Figure 5a*), indicating that slow-cycling chondrocytes in the postnatal growth plate possess a biologically unique pattern of transcriptomes compared to more rapidly diving non-LRCs. Analyses of differentially expressed genes (DEGs) revealed that 799 genes were differentially expressed between the two groups (fold change $\geq \pm 2$), of which 427 and 372 genes were upregulated in LRCs and non-LRCs, respectively (*Figure 5b*, *Supplementary File 1*).

Representative genes upregulated in LRCs included known markers for resting chondrocytes, such as *Pthlh* (also known as *Pthrp*, x2.6) (*Chen et al., 2006*) and *Sfrp5* (x2.4); (*Chau et al., 2014*; *Lui et al., 2010*) in addition to novel markers, such as *Gas1* (x12), *Spon1* (x10), and *Wif1* (x3.8). Similarly, representative genes upregulated in non-LRCs included both known and novel markers for proliferating and pre-hypertrophic chondrocytes, such as *Ihh* (*St-Jacques et al., 1999*) (x54), *Col10a1* (*Gu et al., 2014*) (x11), *Mef2c* (*Arnold et al., 2007*) (x5.1), *Pth1r* (*Hirai et al., 2011*) (x3.0), *Sp7* (*Nakashima et al., 2002*) (x2.4), and *Dlx5* (*Robledo et al., 2002*) (x2.2) as well as *Clec11a* (*Yue et al., 2016*) (x2.9) and *Cd200* (*Etich et al., 2015*) (x2.1). Moreover, a number of genes encoding S and G2/M phase cell cycle regsulators were significantly enriched in non-LRCs, highlighting the

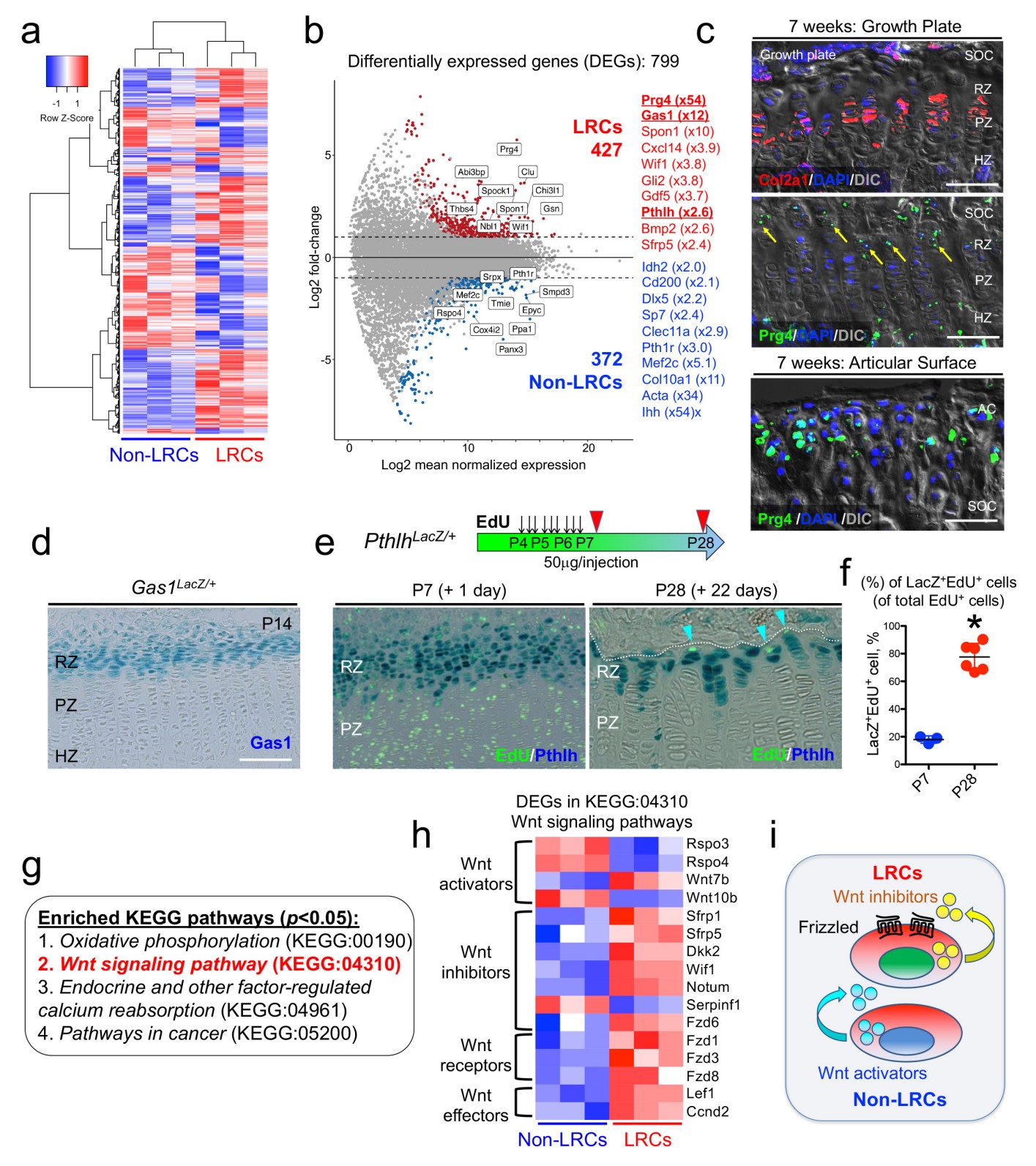

**Figure 5.** Unique molecular signature of label-retaining chondrocytes (LRCs) in the postnatal growth plate. (**a**) Comparative RNA-seq analysis of LRCs and non-LRCs. Heatmap of top 500 differentially expressed genes (DEGs) with hierarchical clustering, between isolated non-LRCs and LRCs. (**b**) MA plot (Log2 fold change) of differentially expressed genes (DEGs) between isolated non-LRCs (372 total) and LRCs (427 total) with representative upregulated genes in each cell population. (**c**) Fluorescent RNAscope in situ hybridization analysis of *Col2a1* and *Prg4* mRNA in growth plate (top) and articular

*Figure 5 continued on next page*

*Figure 5 continued*

surface (bottom) at 7 weeks of age. SOC: secondary ossification center, RZ: resting zone, PZ: proliferating zone, HZ: hypertrophic zone. Blue: DAPI, red: *Col2a1*, green: *Prg4*, gray: DIC. Scale bar 100 μm. (d) *Gas1*$^{LacZ/+}$ distal femur growth plates at P14. Scale bar: 100 μm. (e,f) *Pthlh*$^{LacZ/+}$ distal femur growth plates with EdU administration, serially pulsed nine times between P4 and P6. (e, left panel): Immediately after the pulse at P7. (e, right panel): After 22 days of chase at P28. Arrowheads: EdU label-retaining LacZ$^+$ cells. RZ: resting zone, PZ: proliferating zone. Scale bars: 100 μm. (f): The percentage of LacZ$^+$EdU$^+$ cells among total EdU$^+$ cells, at P7 (*n*=three mice) and P28 (*n*=six mice). *p<0.05, Mann-Whitney's *U*-test. Data are represented as mean ± s.d. (g) Enriched KEGG pathway terms (*p*<0.05) based on 799 differentially expressed genes (DEGs). (h) Heatmap of differentially expressed genes (DEGs) related to KEGG:04310 (canonical Wnt/β-Catenin signaling pathway). The DEGs were further classified by their functions (e.g. Wnt activators, Wnt inhibitors, Wnt receptors and Wnt effectors). (i) Schematic diagram of Wnt activation and inhibition in non-LRCs and LRCs, respectively.

The online version of this article includes the following source data and figure supplement(s) for figure 5:

**Source data 1.** The percentage of Pthlh-LacZ$^+$EdU$^+$cells among total EdU$^+$cells.
**Figure supplement 1.** A comparative RNA-seq analysis of S and G2/M phase cell cycle regulators in LRCs and non-LRCs.
**Figure supplement 2.** An RNAscope analysis of *Col2a1* and *Prg4* expression in postnatal growth plate and articular cartilage.

mitotically active nature of non-LRCs (*Figure 5—figure supplement 1a,b*). Therefore, these identified enriched genes support the precision and accuracy of comparative RNA-seq analysis of LRCs and non-LRCs isolated by cell sorting from the growth plate.

We further set out to validate the LRC markers using several independent approaches. Interestingly, the most highly upregulated gene in LRCs was *Prg4*, which is a marker for chondrocytes in the superficial layer of the articular surface (*Lui et al., 2015*). Chondrocytes on the superficial zone of the articular surface are morphologically similar with those at the resting zone of the growth plate (*Li et al., 2017a*). First, we assessed *Prg4* mRNA expressions in the growth plate and the articular cartilage using RNAscope fluorescent in situ hybridization analysis. At P21, *Prg4* mRNA was indeed found in the resting zone, at a level far fainter than that of the articular surface (*Figure 5—figure supplement 2a*). By 7 weeks, *Prg4* mRNA was also detected in the resting zone in a similar pattern (*Figure 5c*, arrows, *Figure 5—figure supplement 2b*, arrows). Therefore, chondrocytes of the resting zone of the postnatal growth plate express *Prg4* at a low level, supporting the validity of our comparative RNA-seq analysis. Second, we validated expression of a novel LRC marker, *Gas1*. Analysis of *Gas1-LacZ* knock-in mice (*Martinelli and Fan, 2007*) at P14 revealed that Gas1$^+$ cells were exclusively found at the top of the growth plate corresponding to the resting zone (*Figure 5d*). Third, to test if PTHrP$^+$ cells overlap with LRCs, we performed an EdU pulse-chase experiment by serially pulsing *Pthlh*$^{LacZ}$ knock-in mice (*Chen et al., 2006*; *Figure 5e*). Shortly after the pulse at P7, PTHrP$^+$ cells were preferentially localized in an EdU-low zone, wherein 17.9 ± 2.7% of EdU$^+$ cells were PTHrP$^+$ (*Figure 5e*, left panel, and *Figure 5f*). After 22 days of chase at P28, a great majority of EdU-retaining cells were PTHrP$^+$, wherein 77.6 ± 9.6% of EdU$^+$ cells were PTHrP$^+$ (*Figure 5e*, right panel, and *Figure 5f*, ; *Figure 5—source data 1f*). Therefore, LRCs become increasingly enriched among PTHrP$^+$ chondrocytes in the postnatal growth plate. Therefore, in vivo expression patterns of three representative LRC markers – *Prg4*, *Pthlh* and *Gas1* – using in situ hybridization and knock-in reporter lines further support the validity of the gene expression profile of LRCs that accurately reflects that of the resting zone of the growth plate.

Pathway analysis of DEGs revealed significant enrichment of four KEGG terms (*p*<0.05, FDR), including *Oxidative phosphorylation* (KEGG:00190), *Wnt signaling pathway* (KEGG:04310), *Endocrine and other factor-regulated calcium reabsorption* (KEGG:04961) and *Pathways in cancer* (KEGG:05200) (*Figure 5g*). Notably, all DEGs annotated under the *Oxidative phosphorylation* KEGG term were upregulated in non-LRCs, highlighting a biochemically unique feature of non-LRCs undergoing active processes such as cell division and differentiation. Out of 21 DEGs annotated in *Wnt signaling pathway*, 16 genes were relevant to the canonical Wnt/β-catenin signaling pathways (*Komiya and Habas, 2008*; *Figure 5h*). LRCs were enriched for genes encoding Wnt inhibitors such as *Sfrp1*, *Sfrp5*, *Dkk2*, *Wif1*, *Notum* and *Fzd6*, the Wnt activator, *Wnt7b*, and Wnt receptors *Fzd1*, *Fzd3* and *Fzd8*. Conversely, non-LRCs were enriched for genes encoding Wnt activators such as *Rspo3*, *Rspo4* and *Wnt10b* (*Figure 5h*). Therefore, these RNA-seq analyses demonstrate that LRCs reside in a microenvironment in which inhibitors for canonical Wnt signaling are abundantly present in the milieu (*Figure 5i*).

# Activation of canonical Wnt signaling impairs formation, expansion and differentiation of PTHrP$^+$ resting chondrocytes

We next set out to define how canonical Wnt signaling regulates slow-cycling chondrocytes of the postnatal growth plate. For this purpose, we activated Wnt/β-catenin signaling in PTHrP$^+$ resting chondrocytes by conditionally inducing haploinsufficiency of *adenomatous polyposis coli (Apc)*, which is a critical component of the β-catenin degradation complex, using a *Pthlh-creER* (*Mizuhashi et al., 2018*) line and *Apc*-floxed allele (*Cheung et al., 2010*), and simultaneously traced the fates of these Wnt-activated PTHrP$^+$ cells using an R26R-tdTomato reporter allele (*Figure 6a,b*). Littermate triple transgenic mice with two corresponding genotypes – *Pthlh-creER; Apc$^{+/+}$; R26R$^{tdTomato}$* (Control, PTHrP$^{CE}$APC$^{++}$ cells) and *Pthlh-creER; Apc$^{fl/+}$; R26R$^{tdTomato}$* (APC cHet, PTHrP$^{CE}$Apc$^{Het}$ cells) mice – were pulsed with tamoxifen (250 μg) at P6 and analyzed at five consecutive time points after the chase, that is P9, P12, P21 P36, and P96 (*Figure 6c*). Immunohistochemical analysis revealed that the β-catenin protein was significantly upregulated in the resting zone of APC cHet growth plates (CTNNB-Alexa 647 signal: Control, 62.1±33.3; APC cHet: 283.9±65.8, *n*=four mice [p=0.03]), as well as in PTHrP$^{CE}$tdTomato$^+$ cells therein (*Figure 6d*, leftmost panels, arrows, quantification shown in *Figure 6e*, *Figure 6—source data 1e-h*), indicating that *Apc* haploinsufficiency indeed slows β-catenin degradation and activated canonical Wnt signaling specifically in the resting zone of the growth plate.

Subsequently, we quantified the numbers of lineage-marked tdTomato$^+$ cells in the resting zone, as well as short (composed of <10 cells) and long (composed of >10 tdTomato$^+$ cells) columns of tdTomato$^+$ chondrocytes based on serial sections of femur growth plates (*Figure 6d*, right panels, *Figure 6—source data 1e-h*). Consistent with our prior report (*Mizuhashi et al., 2018*), PTHrP$^{CE}$APC$^{++}$ Control chondrocytes transiently increased in the resting zone during the first week of chase and decreased thereafter due to the formation of columnar chondrocytes (P9: 718.7±132.7, P12: 910.3±209.9, P21: 655.4±125.0, P36: 200.3±187.2; P96: 116.1±48.5 cells, *Figure 6f*, blue line, *n*=3–4 mice). In contrast, PTHrP$^{CE}$APC$^{Het}$ resting chondrocytes did not increase in number during the initial stage of chase, the numbers of which were significantly lower than those of Control at the initial three time points (P9: 474.8±134.8 [p=0.04], P12: 558.4±64.3 [p=0.03], P21: 443.4±79.2 [p=0.03] cells, *Figure 6f*, red line, *n*=4–5 mice), and fell to levels that were similar to those in the Control at the latter two time points (*Figure 6d*, rightmost panel, and *Figure 6f*). These data indicate that the formation and the expansion of PTHrP$^+$ cells in the resting zone are slightly impaired when canonical Wnt signaling is activated in these cells.

As expected, PTHrP$^{CE}$APC$^{++}$ resting chondrocytes established short columns (fewer than 10 cells/column) of tdTomato$^+$ chondrocytes across the growth plate that peaked at P21 (P12: 20.0±7.1, P21: 67.4±10.1, P36: 27.5±19.4, P96=44.3±11.1 tdTomato$^+$ columns, *Figure 6g*, blue line, *n*=four mice). The number of tdTomato$^+$ short columns in APC cHet growth plates was reduced at P21 (P21: 45.9±7.7 tdTomato$^+$ columns, *Figure 6g*, red line, *n*=four mice [p=0.03]), indicating that the formation of short columnar chondrocytes in the proliferating zone is inhibited upon canonical Wnt signaling activation. We suspect that this result reflects the reduction of PTHrP-creER$^+$ cells in the resting zone in the preceding stages, although we cannot rule out direct effects of APC haploinsufficiency in the proliferating zone as well. Although we observed significant decreases in columnar chondrocyte formation in APC cHet growth plates, column morphology and organization at P21, P36 and P96 in mutants appear similar compared to Controls (*Figure 6—figure supplement 1*, left, middle, and right panels).

PTHrP$^+$ resting chondrocytes continue to generate long columns (more than 10 cells/column) of tdTomato$^+$ chondrocytes in the long term, the number of which gradually decreases until six months and reaches a plateau thereafter (*Mizuhashi et al., 2018*). Accordingly, PTHrP$^{CE}$APC$^{++}$ cells generated gradually decreasing but still substantial numbers of tdTomato$^+$ long columns after 3 months of chase at P96 (P21: 44.4±23.2, P36: 36.1±34.1, P96: 26.5±12.4 tdTomato$^+$ columns, *Figure 6h*, blue line, *n*=four mice). In contrast, the number of tdTomato$^+$ long columns in APC cHet growth plates was significantly decreased at P96 (P96: 7.3±2.5 tdTomato$^+$ columns, *Figure 6h*, red line, *n*=four mice [p=0.03]). Therefore, the ability for PTHrP$^+$ resting chondrocytes to clonally establish columnar chondrocytes is slightly impaired in response to activation of canonical Wnt signaling in the resting zone.

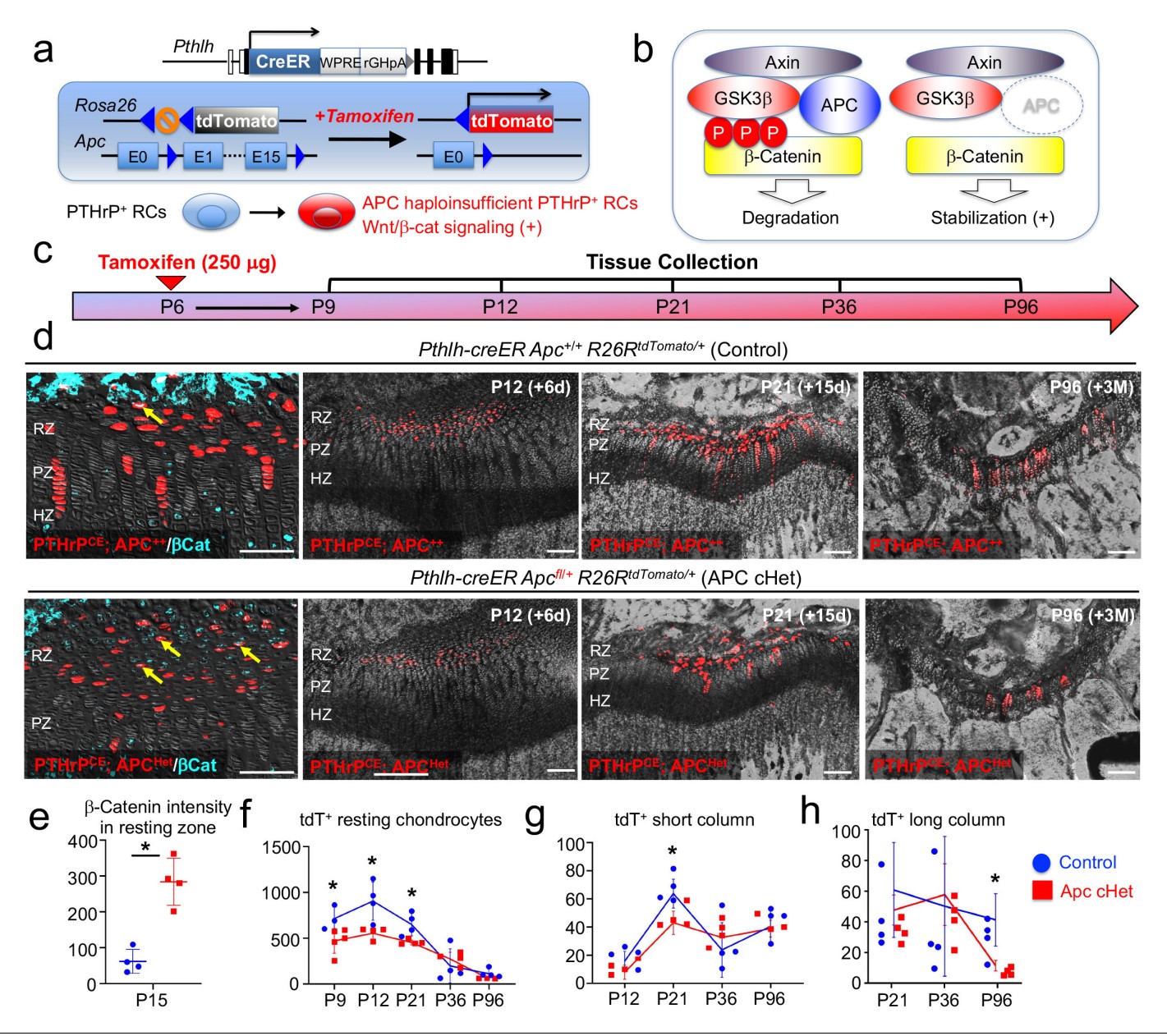

**Figure 6.** Activation of canonical Wnt/β-catenin signaling causes failure of formation and differentiation of PTHrP+ chondrocytes. (a) *Pthlh-creER; R26R^tdTomato* lineage-tracing model crossed with an *adenomatous polyposis coli (Apc)* floxed allele (flanking exons 1 and 15). Single intraperitoneal injection of tamoxifen (0.25 mg) at P6 induces *cre* recombination, leading to activation of canonical Wnt/β-catenin signaling in PTHrP+ chondrocytes via *Apc* haploinsufficiency (*Pthlh-creER; APC^fl/+; R26R-tdTomato*). (b) Schematic diagram of β-catenin degradation complex. Phosphorylation of β-catenin protein leads to degradation (left). *Apc* haploinsufficiency leads to β-catenin stabilization by impairing the degradation complex (right). (c) Timeline for pulse-chase experiment. Tamoxifen injection (0.25 mg) at P6 and chase to P9, P12, P21, P36 and P96. (d) (Leftmost panel): β-catenin staining in *Pthlh-creER; Apc^+/+; R26R^tdTomato* (Control) and *Pthlh-creER; Apc^fl/+; R26R^tdTomato* (APC cHet) distal femur growth plates at P15. Arrows: β-catenin+tdTomato+ cells in RZ. (2nd-4th panels): Distal femur growth plates of *Pthlh-creER; Apc^+/+; R26R^tdTomato* (Control) and *Pthlh-creER; Apc^fl/+; R26R^tdTomato* (APC cHet) at P12, P21, and P96. RZ: resting zone, PZ: proliferating zone, HZ: hypertrophic zone. Blue: β-catenin-Alexa633, red: tdTomato, gray: DAPI and DIC. Scale bars 100 μm. n=four mice per genotype per time point. (e–h) Compiled quantification data of (e) β-Catenin intensity in resting zone (n=four mice per genotype) and total numbers of (f) resting chondrocytes (n=4–5 mice per genotype per time point), (g) short columnar chondrocytes (≤10 tdTomato+ cells, n=four mice per genotype per time point) and (h) long columnar chondrocytes (>10 tdTomato+ cells, n=four mice per genotype per time point) collected from serial sections of femur growth plates (two femurs/mouse) at all time points. *p<0.05, Mann-Whitney's *U*-test. Data are presented as mean ± s.d. Control versus Apc cHet, resting chondrocytes; P9: p=0.036, mean difference = 243.9±97.4, 95% confidence interval (4.2, 483.5); P12: p=0.029, mean difference = 351.9±109.8, 95% confidence interval (83.3, 620.5); P21: p=0.029, mean difference = 198.5±63.9, 95% confidence interval (42.1, 355.0); P36: p=0.343, mean difference = –76.3±100.3, 95% confidence interval (–321.8, 169.3); P96: p=0.057, mean difference = 55.3±28.7,

*Figure 6 continued on next page*

*Figure 6 continued*

95% confidence interval (–18.5, 129.1). Control versus Apc cHet, short columns; P12: p=0.020, mean difference = 7.9±4.3, 95% confidence interval (–2.7, 18.5); P21: p=0.029, mean difference = 20.8±6.5, 95% confidence interval (5.0, 36.5); P36: p=0.343, mean difference = –8.9±10.7, 95% confidence interval (–35.0, 17.3); P96: p=0.343, mean difference = 1.3±7.2, 95% confidence interval (–17.2, 19.7). Control versus Apc cHet, long columns; P21: p=0.886, mean difference = 10.0±12.1, 95% confidence interval (–19.6, 39.6); P36: p=0.686, mean difference = –5.9±18.6, 95% confidence interval (–51.3, 39.5); P96: p=0.029, mean difference = 22.3±6.5, 95% confidence interval (6.2, 38.3).

The online version of this article includes the following source data and figure supplement(s) for figure 6:

**Source data 1.** Compiled quantification data of (e) B-Catenin intensity in resting zone and total numbers of resting chondrocytes, (g) short columnar chondrocytes and (h) long columnar chondrocytes.

**Figure supplement 1.** Morphology and organization of PTHrP[CE] lineage-marked APC-insufficient columnar chondrocytes.

Taken together, these findings indicate that activation of canonical Wnt signaling impairs formation, expansion and differentiation of PTHrP[+] chondrocytes in the resting zone (*Figure 7*). Thus, PTHrP[+] resting chondrocytes are required to be maintained in a Wnt-inhibitory environment to maintain themselves and their column-forming capabilities.

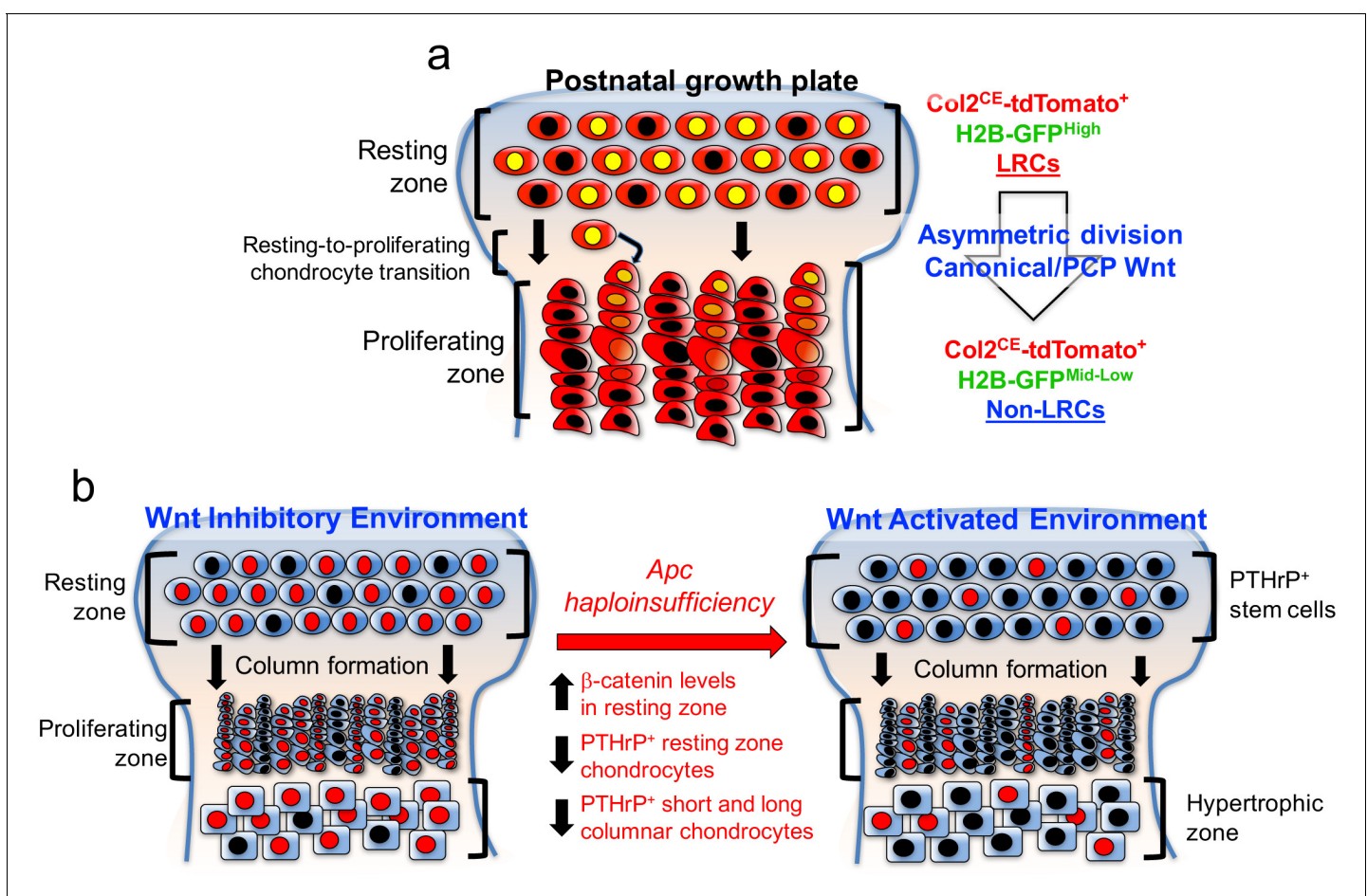

**Figure 7.** PTHrP[+] chondrocytes in the resting zone of the growth plate are maintained in a Wnt-inhibitory environment. PTHrP[+] chondrocytes are maintained in a Wnt inhibitory environment within the resting zone. (a) Transition of chondrocytes from the resting zone ('label-retaining chondrocytes, LRCs') to the proliferating zone ('non-label-retaining chondrocytes, Non-LRCs') in the postnatal growth plate is concertedly regulated by canonical Wnt/β-catenin and non-canonial Wnt/PCP signaling pathways. (b) *Apc* haploinsufficiency increases β-catenin level in the resting zone, and subsequently decreases formation of PTHrP[+] chondrocytes and their differentiation to columnar chondrocytes. Non-canonical Wnt/PCP pathways also play a role in facilitating asymmetric horizontal cell division of resting chondrocytes. Figure Legends – figure supplements.

## Discussion

In this study, we investigated the molecular mechanisms regulating the maintenance and the differentiation of slow-cycling chondrocytes localized in the resting zone of the postnatal growth plate. To date, our understanding of the molecular regulators of this special subclass of chondrocytes is largely grounded in histological and immunohistochemical observations and extrapolations from conditional gene ablation studies (*Hallett et al., 2019*). To address this gap in knowledge, we established a quadruple transgenic murine reporter model, 'Col2-Q' system, to genetically label slow-cycling chondrocytes in an unbiased manner using a pulse-chase approach based on a chondrocyte-specific doxycycline-controllable Tet-Off system regulating expression of histone 2B-linked GFP. We successfully isolated and defined the identities of label-retaining chondrocytes (LRCs) and their proliferating counterparts (non-LRCs), based on combinatorial histological and flow cytometry analyses using a proliferation marker EdU, in order to profile the transcriptome of these cells. As the resting zone of the growth plate is considered to represent a resident stem-cell niche (*Abad et al., 2002*; *Mizuhashi et al., 2018*; *Newton et al., 2019*), our experiments serve as an approach to interrogate the fundamental characteristics of one of the stem-like cells residing in the postnatal growth plate.

It is unclear how slow-cycling chondrocytes in the resting zone maintain low mitotic capabilities while differentiating into columnar chondrocytes in the proliferating zone. Using a comparative bulk RNA-seq transcriptomic analysis, we discovered that LRCs are enriched for a unique set of genes associated with hallmark (e.g. *Pthlh* and *Sfrp5*) and novel (e.g. *Gas1*, *Spon1*, and *Wif1*) markers for resting chondrocytes, in addition to Wnt inhibitory molecules (e.g. *Sfrp1*, *Dkk2*, *Notum*, and *Fzd6*). Interestingly, however, we found that *Wnt7b*, a Wnt activator, is enriched in LRCs. Wnt7b is expressed by perichondral cells adjacent to the hypertrophic zone and simulates differentiation of osteoblasts (*Chen et al., 2014*; *Joeng and Long, 2014*; *Song et al., 2020*) and chondrocytes (*Chen et al., 2014*). Thus, LRCs may be maintained in a delicate balance between Wnt inhibitors and activators. Conversely, non-LRCs were enriched for markers of pre-hypertrophic (e.g. *Ihh*) and hypertrophic (e.g. *Col10a1*) chondrocytes, and represent differentially expressed genes commonly associated with metabolically active cellular processes, such as oxidative phosphorylation. We further validated the expression of *Pthlh*, which is a hallmark marker for resting chondrocytes, and *Gas1*, a novel marker, using *Pthlh-LacZ* and *Gas1-LacZ* knock-in reporter alleles, respectively. We found that PTHrP⁺ chondrocytes in the resting zone maintain low levels of mitotic activity, indicated by EdU labeling and pulse-chase experiments. Thus, the genes identified by our comparative transcriptomic analysis appear to represent accurate transcriptomic features of distinct populations of slow-cycling versus differentiating chondrocytes in the postnatal growth plate. Future investigations aimed at assessing the roles of novel marker genes may lead to the identification of novel skeletal stem cell populations that are important for the postnatal growth plate.

Wnt/β-catenin signaling is essential for endochondral ossification (*Regard et al., 2012*), and is shown to regulate initiation of chondrocyte hypertrophy by inhibiting PTHrP signaling activities (*Guo et al., 2009*). Moreover, Wnt/β-catenin signaling is essential during skeletal development for regulating mesenchymal progenitor differentiation in favor of osteoblasts (*Day et al., 2005*), or for preventing transdifferentiation of osteoblast precursors into chondrocytes (*Hill et al., 2005*). In order to determine the functional contribution of Wnt signaling to PTHrP⁺ resting chondrocyte skeletal stem cells and their differentiation, one copy of *adenomatous polyposis coli* (*Apc*), a critical signaling component of the β-catenin degradation complex, was selectively ablated using a resting chondrocyte-specific *Pthlh-creER* line. In the resting zone, Apc haploinsufficiency led to increased β-catenin protein expression specifically in the resting zone including in PTHrP⁺ chondrocytes, and slightly decreased formation and expansion of PTHrP⁺ chondrocytes, reducing differentiation capabilities of these cells into columnar chondrocytes in the proliferating zone. It is important to note that the effect of β-catenin stabilization in PTHrP⁺ chondrocytes appears to be modest. Lines of studies demonstrate that non-canonical Wnt/planar cell polarity (PCP) signaling plays a key role in facilitating asymmetric divisions of resting chondrocytes. Therefore, our findings support the notion that canonical and non-canonical Wnt signaling pathways concertedly modulate PTHrP⁺ chondrocytes in the resting zone and regulate their differentiation.

Taken together, our data support a novel paradigm that slow-cycling PTHrP⁺ chondrocytes are maintained in a Wnt-inhibitory environment within the resting zone of the growth plate, and that this

relationship is critical to regulating the formation, the expansion and the differentiation of chondrocytes of the resting zone (*Figure 7*).

# Materials and methods

## Key resources table

| Reagent type (species) or resource | Designation | Source or reference | Identifiers | Additional information |
|---|---|---|---|---|
| Strain, strain background (*Mus musculus*) | *Col2a1-tTA* | Jackson Laboratory | RRIDMGI:6490616 | |
| Strain, strain background (*Mus musculus*) | *TRE-H2B-EGFP* | PMID:19060879 | RRIDMGI:5007779 | |
| Strain, strain background (*Mus musculus*) | *Col2a1-creER* | PMID:16894608 | RRIDMGI:3665440 | |
| Strain, strain background (*Mus musculus*) | *Pthlh-LacZ/null* | PMID:16355280 | RRIDMGI:3829430 | |
| Strain, strain background (*Mus musculus*) | *Gas1-LacZ/null* | PMID:11231094 | RRIDMGI:2388397 | |
| Strain, strain background (*Mus musculus*) | *Pthlh-creER* | PMID:30401834 | RRIDMGI:6257033 | |
| Strain, strain background (*Mus musculus*) | *Rosa26-CAG-loxP-stop-loxP-tdTomato* | Jackson Laboratory | RRIDMGI:3809524 | |
| Strain, strain background (*Mus musculus*) | *Apc<sup>tm1Tyj</sup>* | Jackson Laboratory | RRIDMGI:3829069 | |
| Sequence-based reagent | *Mm-Col2a1* | ACDbio | Cat# 407221 | (1:500) |
| Sequence-based reagent | *Mm-Prg4* | ACDbio | Cat# 437661 | (1:500) |
| Antibody | Rabbit polyclonal anti-β-catenin | Abcam | Cat# 16051 | (4 μg) |
| Antibody | Goat polyclonal Alexa Fluor 633-conjugated anti-Rabbit IgG (H+L) | Invitrogen | Cat# A-21071 | (1:200) |
| Antibody | Rat monoclonal APC-conjugated anti-mouse CD45 | Invitrogen | Cat# 17-0451-82 | (1:500) |
| Antibody | Rat monoclonal eFlour450-conjugated anti-mouse CD45 | Invitrogen | Cat# 14-0451-82 | (1:500) |
| Chemical compound, drug | Opal 520 | Akoya Biosciences | Cat# 1601877 | |
| Chemical compound, drug | Alexa Fluor 488-azide | Invitrogen | Cat# A10266 | |
| Chemical compound, drug | Alexa Fluor 647-azide | Invitrogen | Cat# A10277 | |
| Chemical compound, drug | EdU (5-ethynyl-2'-deoxyuridine) | Invitrogen | Cat# E10187 | |

*Continued on next page*

*Continued*

| Reagent type (species) or resource | Designation | Source or reference | Identifiers | Additional information |
|---|---|---|---|---|
| Other | DAPI (4',6-Diamidino-2-Phenylindole, Dihydrochloride) | Invitrogen | Cat# D1306 | |

### Generation of *Col2a1-tTA* transgenic mice

*Col2a1-tTA* transgenic mice were generated by pronuclear injection of a NotI-digested 8.4 kb gene construct containing a 3 kb mouse *Col2a1* promoter and a 3 kb fragment of intron one ligated to a splice acceptor sequence followed by an internal ribosome-entry site (IRES) (*Ovchinnikov et al., 2000*), tetracycline-controlled transactivator (tTA) and the SV40 large T antigen polyadenylation signal (Takara Bio, Mountain View, CA), into B6SJLF1 fertilized eggs. The G0 founder mice were back-crossed with FVB/N mice at least for ten generations. Of the two lines established, the high expresser line (Line H) was used for subsequent studies. The insertion site of the *Col2a1-tTA* transgene was determined based on the Genome Walker Universal system (Takara Bio). The *Col2a1-tTA* transgene was inserted 16kbp downstream of *Pellino2* on Chromosome 14. *Col2a1-tTA* mice were genotyped using PCR primers discriminating heterozygosity and homozygosity of the transgene (85: SV40pA_End_Fw: ACGGGAAGTATCAGCTCGAC, 86: Mm14_5WT_Fw: TTGAGAGTCTCCCAGG-CAAT, 87: Mm14_3WT_Rv: CTCCTGATCTCCTGGCAAAG, ~600 bp for wild-type, ~300 bp for Col2a1-tTA allele). The *Col2a1-tTA* stain is deposited at the Jackson Laboratories (JAX035969).

### Mice

*TRE-H2B-EGFP* (*Foudi et al., 2009*) knock-in, *Col2a1-creER* transgenic (*Nakamura et al., 2006*), *Pthlh-LacZ/null* knock-in (*Chen et al., 2006*), *Gas1-LacZ/null* knock-in (*Martinelli and Fan, 2007*), *Pthlh-creER* transgenic (*Mizuhashi et al., 2018*) mice have been described elsewhere. *Rosa26-CAG-loxP-stop-loxP*-tdTomato (Ai14: *R26R*-tdTomato, JAX007914), *Apc*-floxed (JAX009045) mice (*Cheung et al., 2010*) were acquired from the Jackson Laboratory. All procedures were conducted in compliance with the Guidelines for the Care and Use of Laboratory Animals approved by the University Michigan's Institutional Animal Care and Use Committee (IACUC), protocol 7681 and 9496. All mice were housed in a specific pathogen-free condition, and analyzed in a mixed background. Mice were identified by micro-tattooing or ear tags. Tail biopsies of mice were lysed by a HotShot protocol (incubating the tail sample at 95°C for 30 min in an alkaline lysis reagent followed by neutralization) and used for GoTaq Green Master Mix PCR-based genotyping (Promega, and Nexus X2, Madison, WI). Mice were euthanized by over-dosage of carbon dioxide or decapitation under inhalation anesthesia in a drop jar (Fluriso, Isoflurane USP, VetOne, Boise, ID).

### Doxycycline

Mice were weaned at postnatal day (P) 21 and fed with a standard diet containing 2 mg/g doxycycline (Bio-Serv F3893, Flemington, NJ) for up to 12 weeks.

### Tamoxifen

Tamoxifen (Sigma T5648, St. Louis, MO) was mixed with 100% ethanol until completely dissolved. Subsequently, a proper volume of sunflower seed oil (Sigma S5007) was added to the tamoxifen-ethanol mixture and rigorously mixed. The tamoxifen-ethanol-oil mixture was incubated at 60°C in a chemical hood until the ethanol evaporated completely. The tamoxifen-oil mixture was stored at room temperature until use. Mice with 21 days of age or older received two doses of 2 mg of tamoxifen intraperitoneally at 3 and 2 days prior to analysis, or mice with 6 days of age received a single dose of 0.25 mg tamoxifen intraperitoneally for lineage-tracing analysis.

### Cell proliferation and EdU label-retention assay

5-Ethynyl-2'-deoxyuridine (EdU) (Invitrogen A10044, Carlsbad, CA) dissolved in PBS was administered to mice at indicated postnatal days. Click-iT Imaging Kit with Alexa Flour 488-azide (Invitrogen

C10337) was used to detect EdU in cryosections, or Click-iT Plus Flow Cytometry Kit with Alexa Flour 647-azide (Invitrogen C10634). For *Pthlh-LacZ* EdU label-retention assay, *Pthlh-LacZ* mice received serial doses of EdU (50 μg each) between P4 and P6, and chased for 3 weeks. For Col2-Q EdU proliferation assays, Col2-Q mice received two doses of EdU (500 μg each at P28 and 35, 800 μg each at P49) at 6 and 3 hr prior to sacrifice.

## X-Gal staining of dissected femur epiphyses

Distal epiphyses of femurs were manually dislodged, and attached soft tissues were carefully removed to ensure the maximum penetration of the substrate. Dissected epiphyses were fixed in 2% paraformaldehyde for 30 min at 4°C, followed by overnight X-gal staining at 37°C. Stained samples were further postfixed in 4% paraformaldehyde, overnight at 4°C, then decalcified in 15% EDTA for 7 days. Decalcified samples were cryoprotected in 30% sucrose/PBS followed by 30% sucrose/PBS: OCT (1:1) solution, each overnight at 4°C.

## Histology

Bilateral femurs were dissected under a stereomicroscope (Nikon SMZ-800, Tokyo, Japan) to remove soft tissues, and fixed in 4% paraformaldehyde for a proper period, typically ranging from 3 hr to overnight at 4°C, then decalcified in 15% EDTA for a proper period, typically ranging from 0 hr to 14 days. Decalcified samples were cryoprotected in 30% sucrose/PBS solutions and then in 30% sucrose/PBS:OCT (1:1) solutions, each at least overnight at 4°C. Samples were embedded in an OCT compound (Tissue-Tek, Sakura, Torrance, CA) under a stereomicroscope and transferred on a sheet of dry ice to solidify the compound. Embedded samples were cryosectioned at 14–50 μm using a cryostat (Leica CM1850, Wetzlar, Germany) and adhered to positively charged glass slides (Fisherbrand ColorFrost Plus). Cryosections were stored at −20°C (quantification) or –80°C (immunofluorescence) in freezers until use. Sections were postfixed in 4% paraformaldehyde for 15 min at room temperature. For functional conditional knockout experiments, 50 μm serial sections were collected through the entire growth plate. For immunofluorescence experiments, epiphyses were popped out of bilateral femurs, processed for 24 hr in 4% paraformaldehyde and sectioned at 14 μm. Sections were incubated with anti-β-catenin primary antibody (Abcam ab16051, Cambridge, UK) overnight at 4°C and further stained with 1:200 Alexa Fluor 633 Goat anti-Rabbit IgG (H+L) Secondary Antibody (Invitrogen A21071) at a 20°C for 3 hr. Sections were further incubated with DAPI (4',6-diamidino-2-phenylindole, 5 μg/ml, Invitrogen D1306) to stain nuclei prior to imaging. For EdU assay, sections were incubated with Alexa Fluor 488-azide (*Pthlh-LacZ*, Invitrogen A10266) or Alexa Fluor 647-azide (Col2-Q, Invitrogen A10277) for 30 min at 43°C using Click-iT Imaging Kit (Invitrogen C10337). Sections were further incubated with DAPI to stain nuclei prior to imaging. Stained samples were mounted in TBS with No.1.5 coverslips (Fisher, Waltham, MA).

## RNAscope in situ hybridization

In situ hybridization was performed with RNAscope 2.5 Multiplex Fluorescent V2 Assay (Advanced Cell Diagnostics [Newark, CA, USA] 323100) using the following probes: *Col2a1* (314741) and *Prg4* (437661), according to the manufacturer's fixed frozen tissue protocol. Probes were diluted to 1:500 concentration using Opal 520 reagent (Akoya Biosciences [NC1601877]).

## Imaging and cell quantification

Images were captured by an automated inverted fluorescence microscope with a structured illumination system (Zeiss Axio Observer Z1 with ApoTome.2 system) and Zen 2 (blue edition) software. The filter settings used were: FL Filter Set 34 (Ex. 390/22, Em. 460/50 nm), Set 38 HE (Ex. 470/40, Em. 525/50 nm), Set 43 HE (Ex. 550/25, Em. 605/70 nm), Set 50 (Ex. 640/30, Em. 690/50 nm) and Set 63 HE (Ex. 572/25, Em. 629/62 nm). The objectives used were: Fluar 2.5x/0.12, EC Plan-Neofluar 5x/0.16, Plan-Apochromat 10x/0.45, EC Plan-Neofluar 20x/0.50, EC Plan-Neofluar 40x/0.75, Plan-Apochromat 63x/1.40. Images were typically tile-scanned with a motorized stage, Z-stacked and reconstructed by a maximum intensity projection (MIP) function. Differential interference contrast (DIC) was used for objectives higher than 10x. Some of the images were captured by a fluorescence microscope (Nikon Eclipse E800) with prefigured triple-band filter settings for DAPI/FITC/TRITC, and merged with Spot Advanced Software (Spot Imaging, Sterling Heights, MI). Confocal images were

acquired using LSM510 and Zen2009 software (Zeiss, Oberkochen, Germany) with lasers and corresponding band-pass filters for DAPI (Ex.405nm, BP420-480), GFP (Ex.488nm, BP505-530) and tdTomato (Ex.543nm, BP565-595). LSM Image Viewer and Adobe Photoshop software were used to capture and align images. The number of tdTomato[+] cells, β-Catenin intensity and CTCF values of H2B-GFP were counted by two individuals manually or using ImageJ image analysis software by single blinded methods to ensure unbiased data interpretation.

## Growth plate cell preparation

Distal epiphyses of femurs and proximal epiphyses of tibias were manually dislodged using dull scissors, and attached soft tissues and woven bones were carefully removed using a cuticle nipper. Cells were dissociated from dissected epiphyses using five serial rounds of collagenase digestion, incubating with 2 Wunsch units of Liberase TM (Roche, Basel, Switzerland) in 2 ml $Ca^{2+}$, $Mg^{2+}$-free Hank's Balanced Salt Solution (HBSS, Sigma H6648) at 37°C for 30 min. each time on a shaking incubator (ThermomixerR, Eppendorf, Hamburg, Germany). Single–cell suspension was generated using an 18-gauge needle and a 1 ml Luer-Lok syringe (BD), and filtered through a 70 µm cell strainer (BD) into a 50-ml tube on ice.

## Flow cytometry

Dissociated cells were stained by standard protocols with allophycocyanin (APC)-conjugated anti-mouse CD45 (30 F-11) antibodies (1:500, eBioscience, San Diego, CA). Flow cytometry analysis was performed using a four-laser BD LSR II (Ex. 355/407/488/633 nm) or LSR Fortessa (Ex. 405/488/561/640 nm) and FACSDiva software. Acquired raw data were further analyzed on FlowJo software (TreeStar). Representative plots of at least three independent biological samples are shown in the figures.

## Fluorescence-activated cell sorting (FACS) and RNA isolation

Cell sorting was performed using a five-laser BD FACS Aria III (Ex.355/407/488/532/633 nm) and FACSDiva. CD45[neg]GFP[high] cells at the top 10 percentile of GFP brightness (LRCs) and CD45[neg]GFP-[mid-low] cells with 10~50 percentile of GFP brightness (non-LRCs) were directly sorted into TRIzol LS Reagent (ThermoFisher 10296010, Waltham, MA). Total RNA was isolated using NucleoSpin RNA XS (Macherey-Nagel, 740902). RNA Integrity Number (RIN) was assessed by Agilent 2100 Bioanalyzer RNA 6000 Pico Kit. Samples with RIN>8.0 were used for subsequent analyses.

## RNA amplification and deep sequencing

Complementary DNAs were prepared by SMART-Seq v4 Ultra Low Input RNA Kit for Sequencing (Takara 634888) using 150~800 pg of total RNA. Post-amplification quality control was performed by Agilent TapeStation DNA High Sensitivity D1000 Screen Tape system. DNA libraries were prepared by Nextera XT DNA Library Preparation Kit (Illumina) and submitted for deep sequencing (Illumina HiSeq 2500).

## RNA-seq analysis

cDNA libraries were sequenced using following conditions; six samples per lane, 50 cycle single end read. Reads files were downloaded and concatenated into a single. fastq file for each sample. The quality of the raw reads data for each sample was checked using FastQC to identify quality problems. Tuxedo Suite software package was subsequently used for alignment (using TopHat and Bowtie2), differential expression analysis, and post-analysis diagnostics. FastQC was used for a second round of quality control (post-alignment). HTSeq/DESeq2 was run using UCSC mm10.fa as the reference genome sequence. Expression quantitation was performed with HTSeq, to count non-ambiguously mapped reads only. HTSeq counts per gene were then used in a custom DESeq2 paired analysis. Normalization and differential expression were performed with DESeq2, using a negative binomial generalized linear model, including a term for mouse of origin for a paired analysis. Plots were generated using variations or alternative representations of native DESeq2 plotting functions, ggplot2, plotly, and other packages within the R environment. Heatmaps were generated with updated rlog normalized count values for each sample for all plus top sets (500) of differentially expressed genes with the gplots package (v 3.0.1). Two types of clustering were used: (1) averaging

across rows with Pearson correlation distance with average linkage and (2) Ward's squared dissimilarity criterion. Top differentially expressed genes were determined after ranking genes by standard deviation across all samples. Independent of iPathway, GO term enrichment was performed on DE results, with a logFC threshold of 2 and adjusted p-value < 0.05 with the GOseq package (v 1.36) with probability weighting function and GO enrichment specified with mm10 as genome and gene symbol specified as gene ID format. Results were plotted for the top ten of selected terms related to the Wnt pathways, ranked by overrepresented p-value using ggplot2 (v 3.2.1). KEGG results with FDR correction and gene tables for Wnt signaling pathway were downloaded from iPathway (report ID: 41865). KEGG gene tables for each pathway were used to subset the DE results before restricting results to genes for which both log fold change and adjusted p-value statistics were available.

## Replicates

All experiments were performed in biological replicates. For all data presented in the manuscript, we examined at least three independent biological samples (three different mice) to ensure the reproducibility. Biological replicates were defined as multiple experimental samples sharing common genotypes and genetic backgrounds. For each series of the experiments, all attempts at biological replication were successful. Technical replicates were generated from a single experimental sample. For example, serial sections of the femur growth plate from a single mouse were considered technical replicates. Outliers were uncommon in our datasets and did not impact the trend and the significance of our quantitated results. As a result, all quantitative data were included to ensure transparency in our data interpretation.

## Statistical analysis

Results are presented as mean values ± s.d. Statistical evaluation was conducted based on Mann-Whitney's *U*-test. A p value <0.05 was considered significant. No statistical method was used to predetermine sample size. Sample size was determined on the basis of previous literature and our previous experience to give sufficient standard deviations of the mean so as not to miss a biologically important difference between groups. The experiments were not randomized. All the available mice of the desired genotypes were used for experiments. The investigators were not blinded during experiments and outcome assessment. One femur from each mouse was arbitrarily chosen for histological analysis. Genotypes were not particularly highlighted during quantification.

## Acknowledgements

This research was supported by grants from National Institute of Health (R01DE026666 and R01DE030630 to NO, R01DE029181 to WO, P01DK011794 to HMK and T32DE007057 to SAH). We thank H Hock for *TRE-H2B-EGFP* mice and B Allen for *Gas1-LacZ* mice. We acknowledge support from the Bioinformatics Core of the University of Michigan Medical School's Biomedical Research Core Facilities.

## Additional information

### Funding

| Funder | Grant reference number | Author |
|---|---|---|
| National Institute of Dental and Craniofacial Research | R01DE026666 | Noriaki Ono |
| National Institute of Dental and Craniofacial Research | R01DE030630 | Noriaki Ono |
| National Institute of Dental and Craniofacial Research | R01DE029181 | Wanida Ono |
| National Institute of Diabetes and Digestive and Kidney Diseases | P01DE011794 | Henry M Kronenberg |
| National Institute of Dental and Craniofacial Research | T32DE007057 | Shawn A Hallett |

The funders had no role in study design, data collection and interpretation, or the decision to submit the work for publication.

## Author contributions

Shawn A Hallett, Conceptualization, Data curation, Formal analysis, Funding acquisition, Validation, Investigation, Visualization, Writing - original draft, Writing - review and editing; Yuki Matsushita, Data curation, Formal analysis, Investigation, Methodology; Wanida Ono, Conceptualization, Resources, Data curation, Funding acquisition, Investigation, Methodology, Writing - review and editing; Naoko Sakagami, Conceptualization, Data curation, Formal analysis, Investigation, Visualization; Koji Mizuhashi, Conceptualization, Data curation, Formal analysis, Investigation, Visualization, Methodology; Nicha Tokavanich, Data curation, Investigation, Visualization; Mizuki Nagata, Data curation, Visualization, Methodology; Annabelle Zhou, Data curation, Formal analysis, Visualization; Takao Hirai, Conceptualization, Resources, Data curation, Methodology; Henry M Kronenberg, Conceptualization, Resources, Supervision, Funding acquisition, Project administration, Writing - review and editing; Noriaki Ono, Conceptualization, Resources, Data curation, Formal analysis, Supervision, Funding acquisition, Validation, Investigation, Visualization, Methodology, Writing - original draft, Project administration, Writing - review and editing

## Author ORCIDs

Shawn A Hallett (D) https://orcid.org/0000-0003-1472-7502
Noriaki Ono (D) https://orcid.org/0000-0002-3771-8230

## Ethics

Animal experimentation: All procedures were conducted in compliance with the Guidelines for the Care and Use of Laboratory Animals approved by the University Michigan's Institutional Animal Care and Use Committee (IACUC), protocol 7681 and 9496.

## Decision letter and Author response

Decision letter https://doi.org/10.7554/eLife.64513.sa1
Author response https://doi.org/10.7554/eLife.64513.sa2

# Additional files

## Supplementary files

• Supplementary file 1. Normalized expression count values of label-retaining chondrocytes (LRCs) and non-LRCs. Comparative RNA-seq analysis of LRCs and non-LRCs, rlog normalized count values for each biological sample of LRCs (High1_50032, High2_53561, High3_53563) and non-LRCs (Low1_50033, Low2_53562, Low3_53564). Provided is a table of differential expression (DE) statistics, rlog counts per sample, as well as standard deviation and mean across sample counts in rlog scale. This table was used for generating all heatmaps, including restricting only to differentially expressed genes, to input DE statistics for GO term enrichment, and intersected with KEGG gene sets from iPathway Guide for STRING analysis.

• Transparent reporting form

## Data availability

The bulk RNA-seq datasets presented herein have been deposited in the National Center for Biotechnology Information (NCBI)'s Gene Expression Omnibus (GEO), and are accessible through GEO Series accession number GSE160364 (https://www.ncbi.nlm.nih.gov/geo/query/acc.cgi?acc=GSE160364); rlog normalized expression counts of the RNA-seq datasets are provided in Supplementary file 1. The source data underlying all Figures and Figure supplements are provided as a Source data files. All the raw images and flow cytometry files supporting the conclusion of this study have been deposited on Dryad and are accessible via https://doi.org/10.5061/dryad.70rxwdbz1.

The following datasets were generated:

| Author(s) | Year | Dataset title | Dataset URL | Database and Identifier |
|---|---|---|---|---|
| Ono N | 2020 | Comparative bulk RNAseq analysis of label-retaining chondrocytes (LRCs) of the growth plate versus their progeny | https://www.ncbi.nlm.nih.gov/geo/query/acc.cgi?acc=GSE160364 | NCBI Gene Expression Omnibus, GSE160364 |
| Ono N | 2020 | Chondrocytes in the resting zone of the growth plate are maintained in a Wnt-inhibitory environment | https://doi.org/10.5061/dryad.70rxwdbz1 | Dryad Digital Repository, 10.5061/dryad.70rxwdbz1 |

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
