## [Decision Letter]

**Acceptance summary:**

The resting zone of the epiphyseal growth plate plays a critical role in maintaining the growth plate. There remains a great deal not known about how the cells of the resting zone are maintained and what signals differentiation of these slow dividing cells. In this exceedingly innovative study, the authors elegantly use a complex set of genetic alleles in mice to label and trace maturation of these cells and to elucidate the molecular mechanisms that maintain their stem cell properties/microenvironment. They couple this work with bulk RNA seq and determine that canonical WNT signaling plays role in the maintenance and differentiation of the resting zone chondrocytes. The two major weaknesses are that questions remain for whether their approach can truly mark only the LRCs (as they call them), and the fact that WNT signaling inhibition being important in the resting zone is not entirely novel. The reviewers were all very enthusiastic about this brilliantly conducted study and strongly felt that these weaknesses could be overcome in a reasonable time frame.

**Decision letter after peer review:**

Thank you for submitting your article "Chondrocytes in the resting zone of the growth plate are maintained in a Wnt-inhibitory environment" for consideration by *eLife*. Your article has been reviewed by 3 peer reviewers, and the evaluation has been overseen by a Reviewing Editor and Kathryn Cheah as the Senior Editor. The reviewers have opted to remain anonymous.

The reviewers have discussed the reviews with one another and the Reviewing Editor has drafted this decision to help you prepare a revised submission.

Summary:

The resting zone of the epiphyseal growth plate plays a critical role in maintaining the growth plate. There remains a great deal not known about how the cells of the resting zone are maintained and what signals differentiation of these slow dividing cells. In this exceedingly innovative study, the authors elegantly use a complex set of genetic alleles in mice to label and trace maturation of these cells and to elucidate the molecular mechanisms that maintain their stem cell properties/microenvironment. They couple this work with single cell RNA seq and determine that canonical WNT signaling plays role in the maintenance and differentiation of the resting zone chondrocytes. The two major weaknesses are that questions remain for whether their approach can truly mark only the LRCs (as they call them), and the fact that WNT signaling inhibition being important in the resting zone is not entirely novel. The reviewers were all very enthusiastic about this brilliantly conducted study and strongly felt that these weaknesses could be overcome in a reasonable time frame.

Essential revisions:

1. Line 59-60: The authors stated that "mechanisms regulating self-renewal and differentiation capabilities of resting zone chondrocytes remain largely unknown." That is not entirely true. Published in vivo evidence suggests that Gsa abd Gq/11a signaling contributes to the maintenance of resting cells (Chagin, 2014 Nat Commun). Please consider the strengths and weaknesses of this prior work in the intro of this paper.

2. The authors used an interesting approach of combining the Col2-tTA TRE-H2B-GFP and the Col2-creER R26-Tomato lines. The reviewers were curious as to why the Col2-creER R26-Tomato did not label the chondrocytes in the articular cartilage as these cells also express Col2a1. Would not some (even though not a great deal) of the articular chondrocytes will also have a green nuclei and a red cytoplasm as a result?

3. Line 146-148: The authors stated that "…more rapidly dividing and morphologically distinct columnar chondrocytes were not marked by H2B-EGFP signal. Therefore, our Col2-Q quadruple transgenic strategy can effectively mark LRCs primarily in the resting zone of the postnatal growth plate." But as can be seen in figure 1e, it's not true that the divided cells do not have H2B-GFP. The cells right below the resting zone, which have presumably divided once, have slightly weakened H2B-GFP (showing a yellow nuclei) and the cell below it have yet weaker H2B-GFP. Judging by that figure 1e alone, it seems the H2B-GFP will mark the LRC and then to a certain extent the few cells right below. At the very least, figure 1c and 2b seem somewhat misleading. While choosing the top 10% brightness is somewhat common when isolating high GFP cells, in this specific study, there is more that needs to be taken into account. It is not clear how the authors can know for sure that the GFP high cells they are identifying as LRCs do not also contain cells right below the resting zone that have undergone 1 division or maybe even 2 divisions. More information is needed to clarify what cells are being isolated by flow and the reviewers agreed that a proliferation marker could be helpful in this context. As can be seen in figure 1 and 2 (like figure 2a), the cut off for LRCs vs non-LRCs is at best arbitrary.

4. Line 157-158: The authors stated that "Five rounds of digestion completely liberated cells from the growth plate, while cells on the articular surface were almost undigested" and this statement is supported by the image in Figure 1f. It is unclear why or if collagenase digestion cannot or did not liberate the articular chondrocytes. On close examination, the appearance (focus on the shape of the edge) of the articular cartilage also changed after digestion (even though judging from the SOC those two are probably not the same section). I suspect that some cells in the superficial zone of the articular surface might also have come off and if those cells would be included in downstream experiments like RNA-Seq. Although it looks from figure 1d that the articular cartilage is not labeled either red or green, but the superficial zone of the articular cartilage possesses cells that are quite similar to the resting zone (see for example Li, FASEB J. 2017) and, at least in theory, are those kinds of cells would retain the GFP-H2B. Interestingly, the most differentially expressed gene in RNA-Seq is Prg4, a well-known superficial zone marker but largely undetected in the resting zone (see Lui, 2015 Ped Res).

5. Line 169-171: the authors stated "Therefore, these findings demonstrate that LRCs can be effectively identified and isolated from postnatal Col2-Q growth plates by combined microdissection, enzymatic digestion and flow cytometry-based approaches". It appears what the authors are showing is that chasing by dox removal, over time, is effective in removing GFP signaling in the digested cell population. However, whether the LRCs can be effectively identified and isolated depends on how much GFP-H2B signal diminishes after 1 (or more) cell division and perhaps other factors. Please address this issue.

6. The efforts by the authors to identify and isolate the LRCs population and to perform RNA-Seq to assist with this effort were appreciated by the reviewers. The impact of technical factors (dissection, digestion, label, flow cytometry, etc) and the generalized handling of the cells could potentially affect the fidelity of gene expression study. This is not a situation unique to this study. How long after the growth plate was taken until the RNA were eventually isolated from the cells? What measures where taken to control for or account for these possible technical sources of artifactual signals?

7. In their RNA-Seq experiment, the authors defined non-LRCs as the "GFP mid-low cells". What cells are these in terms of histological position in the growth plate? Are these mostly proliferative chondrocytes or includes the entirety of the PZ-PHZ-HZ cells? The expression profile seems to indicate these could a bit of everything (high in ColX, Ihh, Dlx5, etc). It would have been even more interesting if the non-LRCs are the direct progenies of the LRCs (like the one cell that just exited the resting zone).

8. In figure 3, the authors again utilized a nice combination of mouse models to test how canonical WNT signaling affects resting chondrocytes. Based on the results from figure 3d-g, it seems the effect is quite subtle, which is slightly disappointing. The conclusions from this find are described in an over zealous manner and should be tempered (e.g. "slightly impaired" in line 262).

9. The fact that the authors identified genes like Rspo3, Sfrp5, and Dkk2, suggests WNT signaling but does not automatically mean canonical wnt/catenin signaling. There is a reasonable amount of literature connecting these WNT modulators to other non-canonical WNT, such as the PCP WNT pathway (e.g. Ohkawara, 2011 Dev Cell). In particular, PCP WNT pathway have been implicated in resting zone chondrocyte transition to the proliferative zone (e.g. Andrew Dudley's work in 2009 Development, Tracy Ballock's work in 2012 J Orth Res, and more recently Li, 2017 in *eLife*). It is understood that PCP WNT signaling is activated when the chondrocytes exit the resting zone and start forming columns, thus it is not surprising at all that a lot of WNT modulator/inhibitors are highly expressed in the resting zone. The statement in line 330 "no previous report ties WNT signaling to the maintenance of putative skeletal stem cell populations in the resting zone of the growth plate" is erroneous. While the authors have advanced the field (particularly testing how activating canonical wnt might impact resting zone in figure 3), it is not appropriate to ignore or downplay the prior literature in this area.

10. Figure 3d and lines 245 to 248: The image is unlikely enough to support their claim that b-catenin protein is upregulated in PTHrP-CE; APC-het. The authors are encouraged to show quantitative data for the claim.

---

## [Author Response]

Summary:The resting zone of the epiphyseal growth plate plays a critical role in maintaining the growth plate. There remains a great deal not known about how the cells of the resting zone are maintained and what signals differentiation of these slow dividing cells. In this exceedingly innovative study, the authors elegantly use a complex set of genetic alleles in mice to label and trace maturation of these cells and to elucidate the molecular mechanisms that maintain their stem cell properties/microenvironment. They couple this work with single cell RNA seq and determine that canonical WNT signaling plays role in the maintenance and differentiation of the resting zone chondrocytes. The two major weaknesses are that questions remain for whether their approach can truly mark only the LRCs (as they call them), and the fact that WNT signaling inhibition being important in the resting zone is not entirely novel. The reviewers were all very enthusiastic about this brilliantly conducted study and strongly felt that these weaknesses could be overcome in a reasonable time frame.

Thank you very much for providing this explicit editorial summary. In the revised manuscript, we have addressed the two major weaknesses that were pointed out.

First, we have now better defined the identity of label-retaining chondrocytes (LRCs), by performing additional histological and flow cytometry experiments using a proliferation marker, EdU. These analyses demonstrated that LRCs by our definition are relatively resistant to EdU incorporation and localized to the resting zone. Although these LRCs are still heterogeneous and include cascades of resting chondrocytes, we believe that these cells represent a biologically unique group of growth plate chondrocytes.

Second, we have now extensively discussed the existing literature on the essential role of non-canonical PCP/WNT pathways in the transition of resting chondrocytes to proliferating counterparts and put our study in the context of these studies.

Please see below for the detailed point-to-point responses to the reviewers’ queries.

Essential revisions:1. Line 59-60: The authors stated that "mechanisms regulating self-renewal and differentiation capabilities of resting zone chondrocytes remain largely unknown." That is not entirely true. Published in vivo evidence suggests that Gsa abd Gq/11a signaling contributes to the maintenance of resting cells (Chagin, 2014 Nat Commun). Please consider the strengths and weaknesses of this prior work in the intro of this paper.

We would like to apologize for missing this important reference in the original manuscript. We have now discussed this work in the Introduction.

The following section has been added to the Introduction:

“G protein stimulatory subunit-α (G_s_α), G_q_/G_11_α G proteins, which are coupled with the PTH/PTHrP receptor (PPR), are both required for maintaining quiescent stem-like chondrocytes (Chagin et al., 2014). Pan-chondrocyte ablation of G_s_α (Col2a1-creER; Gnas^f/f^) causes premature differentiation of stem-like resting chondrocytes into the proliferative pool, resulting in accelerated endochondral bone growth. Further, combined inactivation of G_q_/G_11_α through a mutant PPR (Guo et al., 2002) and G_s_α (Col2a1-creER, Gnas^f/f^; PPR^D/D^) causes a more severe phenotype associated with growth plate fusion. Therefore, PPR-mediated G_s_α and G_q_/G_11_α synergistically maintain quiescence of resting chondrocytes and their differentiation into columnar chondrocytes (Chagin et al., 2014); however, whether this regulation occurs cell-autonomously in resting chondrocytes has not been determined.”

2. The authors used an interesting approach of combining the Col2-tTA TRE-H2B-GFP and the Col2-creER R26-Tomato lines. The reviewers were curious as to why the Col2-creER R26-Tomato did not label the chondrocytes in the articular cartilage as these cells also express Col2a1. Would not some (even though not a great deal) of the articular chondrocytes will also have a green nuclei and a red cytoplasm as a result?

Thank you very much for pointing out this important topic for discussion. As demonstrated in the new Figure 5 —figure supplement 2, and as the reviewers pointed out, *Col2a1* mRNA is indeed expressed by chondrocytes in the articular cartilage, both at P21 and 7 weeks of age, although its expression level dwindles in the later stage.

However, interestingly, *Col2a1-creER* does not appear to mark chondrocytes in the articular cartilage in adulthood. This point has been previously reported by Nagao et al., where the authors discovered that the labeling of articular chondrocytes by *Col2a1-creER* dwindles during the postnatal period. Tamoxifen administration into *Col2a1-creER; R26R^tdTomato^* mice at P14, P21, P28 and P56 results in 90.8%, 46.0%, 22.2% and little to no tdTomato^+^ cells, respectively, in the articular surface. We think that our findings that *Col2a1-creER* does not mark articular chondrocytes are in line with this previous report.

To further address the reviewers’ concerns, we counted the number of Col2a1^CE^-tdTomato^+^ cells in the growth plate and the articular cartilage using serial sections. As shown in new Figure 3b, *Col2a1-creER* marked a far greater number of chondrocytes in the growth plate than in the articular cartilage at P49. Further, as shown in new Figure 2e, left, we observed a number of H2B-GFP^high^ chondrocytes in the articular cartilage, but essentially none of these cells were also labeled with Col2a1^CE^-tdTomato.

Therefore, the use of a *Col2a1-creER* transgenic line allows us to almost specifically label growth plate chondrocytes, when recombination is induced at postnatal stages (~7 weeks of age).

The following sentences have been added to the Results:

“A great majority of cells in the epiphysis, including those in the growth plate and the secondary ossification center, but not as many on the articular surface, were H2B-EGFP^high^ (Figure 2d, cells with green nuclei, Figure 2e, left).”

The following section has been added to the Results:

“First, we quantified Col2a1^CE^-tdT^+^ cells at P49 with tamoxifen injection at 2 and 3 days prior to dissection. Importantly, we observed a significantly fewer number of Col2a1^CE^-tdT^+^ chondrocytes in the articular surface compared to those in the growth plate at P49 (Figure 3a,b; GP=1,315.0±171.2 cells; AC=79.1±50.3 cells, n=9 mice [p<0.0001]), consistent with the previous finding that Col2a1-creER preferentially marks growth plate chondrocytes in adulthood (Nagao et al., 2016). Therefore, our short-chase tamoxifen protocol with Col2a1-creER enables preferential labeling of growth plate chondrocytes in adulthood.”

The following section has been added to the Results:

“We further sought to determine whether collagenase digestion enables the preferential isolation of growth plate chondrocytes. To this end, we quantified Col2a1^CE^-tdT^+^ cells in the growth plate and articular surface before and after collagenase digestion at P49 (Figure 3b). We found a significant reduction in Col2a1^CE^-tdT^+^ chondrocytes in the growth plate after collagenase digestion (Before: GP=1,315.0±171.2, After: GP=49.8±45.2, n=9 [before], n=4 [after] mice [p=0.003]). In contrast, there was no change in the number of Col2a1^CE^-tdT^+^ chondrocytes in the articular cartilage following collagenase digestion (Before: AC=79.1±50.3, After: AC=85.8±45.2, n=9 [before], n=4 mice [after] [p=0.939]). Based on this data, we further enumerated the percentage of growth plate chondrocytes among total Col2a1^CE^-tdT^+^ cells dissociated from dissected the epiphyses. In fact, growth plate chondrocytes accounted for essentially all of Col2a1^CE^-tdT^+^ cells (99.1±1.4% of Col2a1^CE^-tdT^+^ cells, n=4 mice). Therefore, these data demonstrate that our microdissection and enzymatic dissociation approach allows us to selectively harvest Col2a1^CE^-tdT^+^ chondrocytes from the postnatal growth plate.”

3. Line 146-148: The authors stated that "…more rapidly dividing and morphologically distinct columnar chondrocytes were not marked by H2B-EGFP signal. Therefore, our Col2-Q quadruple transgenic strategy can effectively mark LRCs primarily in the resting zone of the postnatal growth plate." But as can be seen in figure 1e, it's not true that the divided cells do not have H2B-GFP. The cells right below the resting zone, which have presumably divided once, have slightly weakened H2B-GFP (showing a yellow nuclei) and the cell below it have yet weaker H2B-GFP. Judging by that figure 1e alone, it seems the H2B-GFP will mark the LRC and then to a certain extent the few cells right below. At the very least, figure 1c and 2b seem somewhat misleading. While choosing the top 10% brightness is somewhat common when isolating high GFP cells, in this specific study, there is more that needs to be taken into account. It is not clear how the authors can know for sure that the GFP high cells they are identifying as LRCs do not also contain cells right below the resting zone that have undergone 1 division or maybe even 2 divisions. More information is needed to clarify what cells are being isolated by flow and the reviewers agreed that a proliferation marker could be helpful in this context. As can be seen in figure 1 and 2 (like figure 2a), the cut off for LRCs vs non-LRCs is at best arbitrary.

Thank you for these critical comments and recommendations. We agree with the reviewers that a better definition of label-retaining chondrocytes (LRCs) is essential to sustaining the conclusions of our study. The important limitation of our study is that the transition between LRCs and non-LRCs are almost always contiguous. We therefore needed to define LRCs with somewhat arbitrary criteria in the original manuscript: (1) histologically, we defined LRCs as H2B-EGFP^high^ cells at the top of the growth plate, and (2) by flow cytometry, we defined LRCs as cells with the top 10% brightness.

To support the authenticity of these criteria, we have performed additional experiments to justify the LRC term in our study.

First, we have quantified the intensity of H2B-EGFP signal through the entire column of chondrocytes using the corrected total cell fluorescence (CTCF) after two weeks of chase at P35, as shown in new Figure 1b,c.

This quantitative data support the reviewers’ statement that the cells right below the resting zone have slightly less H2B-EGFP signal, and the cells below them have yet weaker H2B-EGFP signal. However, we also confirmed that the cells with the top 10% brightness corresponded to resting chondrocytes at the top of the growth. While there were 27 cells in this given column of chondrocytes, the 3 brightest cells (374,012; 343,652; 264,697) were localized to the resting zone (Figure 1b, middle panel, arrows). Thus, based on this quantitative histological data, we believe that the top 10% brightness is a reasonable threshold to identify LRCs. In the revised manuscript, we have updated the description to clarify the definition of LRCs and non-LRC.

The following section has been added to the Results:

“Next, to more rigorously define LRCs and non-LRCs in the postnatal growth plate, we quantified the intensity of H2B-EGFP signal through the entire column of chondrocytes using the corrected total cell fluorescence (CTCF) after two weeks of doxycycline treatment at P35 in double homozygous Col2a1-tTA/Col2a1-tTA; TRE-H2B-GFP/TRE-H2B-EGFP mice. We labelled actively proliferating chondrocytes by pulsing these mice with a thymidine analogue EdU twice at 6 and 3 hours prior to sacrifice. By this approach, we expect to visualize an EdU-positive cell in the proliferating zone that is descended from an EdU-negative cell in the resting zone through one cell division (Figure 1b, left panel, arrows). The quantification of CTCF values through the entire column revealed that the cells in the proliferating zone (PZ) had weaker H2B-EGFP signal than those of the resting zone (RZ), and the cells in the pre-hypertrophic (PHZ) and hypertrophic zones (HZ) had yet weaker H2B-EGFP signal. The side-by-side comparison of CTCF values between EdU^+^ cells in the proliferating zone and their preceding EdU-negative cells in the resting zone further revealed that H2B-EGFP CTCF values decreased significantly upon cell division (Figure 1c, P35: CTCF, EdU^-^ cells in RZ = 267,184.55±109,457.82; EdU^+^ cells in PZ = 131,891.99±83,344.76, n=4 mice, 24 data points [p<0.0001]). The average fold change between EdU-negative and EdU-positive cells was 2.03, suggesting that H2B-EGFP intensity decreases by a factor of two following one cell division (Figure 1b,c), as expected from the fact that histone-bound GFP is distributed equally between the two daughter cells upon cell division. Further, the cells with the top 10% brightness were localized to RZ at the top of the growth plate; while there were 27 cells in this given column of chondrocytes, the 3 brightest cells (374,012; 343,652; 264,697) were localized to RZ (Figure 1b, arrowheads). Thus, based on these quantitative histological data, we designated the cells with the top 10% brightness as LRCs.”

The following underlined sentences have been added to the Results:

**“**After the chase, LRCs were identified at a specific location near the top of the growth plate in the resting zone, retaining a higher level of H2B-EGFP signal (Figure 2e, right panel). In addition, most of these H2B-EGFP^high^ cells in the growth plate were simultaneously marked as Col2^CE^-tdT^+^ (Figure 2e, right panel, arrowhead, Figure 2c). While chondrocytes with the brightest H2B-GFP signal were localized to the resting zone, their descendants showed increasingly diluted H2B-EGFP signals as they progressed toward the proliferating and pre-hypertrophic zones (Figure 1b, Figure 2e, right panel).”

Second, following the reviewers’ directives, we have performed additional flow cytometry experiments using a proliferation marker EdU, to clarify the identity of LRCs and non-LRCs that are isolated for the downstream RNA-seq analysis. To this end, we pulsed Col2-Q mice with two serial doses of the thymidine analogue, EdU, at 3 and 6 hours prior to sacrifice. Under this protocol, actively proliferating cells should incorporate EdU. We used Click-iT Plus Alexa 647 Flow Cytometry kit, to fluorescently label EdU-incorporated cells. We next performed flow cytometry analysis of LRCs as cells with the top 10% H2B-EGFP brightness and non-LRCs as cells with 10-50% units of H2B-GFP brightness. We quantified the percentage of cells positive for EdU-Alexa Fluor 647 among LRCs and non-LRCs.

We observed that non-LRCs were significantly enriched for EdU^+^ cells, both at 1 week and 2 weeks of chase (new Figure 4d, new Figure 4e, left and middle and new Figure 4 —figure supplement S2a). We also found a trend that non-LRCs were enriched for EdU^+^ cells at 4 weeks of chase, although this was not statistically significant (new figure 4e, right and new Figure S2b). Therefore, LRCs are relatively resistant to EdU incorporation, denoting their relatively quiescent status.

Third, we examined the expression levels of cell cycle regulator genes in our comparative RNA-seq analysis, as shown in the new Figure S4. A number of S and G2/M phase cell cycle regulator genes were differentially expressed between LRCs and non-LRCs, with positive regulators particularly enriched in non-LRCs supporting the notion that non-LRCs are preferentially in cell division.

In conclusion, based on the additional quantitative data that we presented, we strongly believe that the cells with the top 10% H2B-EGFP brightness is a reasonable definition of LRCs, because these cells are (1) histologically localized to the resting zone and (2) resistant to incorporation of a proliferation marker, EdU, and negatively enriched for cell cycle regulator genes.

We acknowledge that these LRCs are still heterogeneous and perhaps include early cascades of resting chondrocytes. However, we believe that these LRCs represent a biologically unique group of growth plate chondrocytes.

The following section was added to the Results:

“We next set out to define whether LRCs isolated in single-cell suspension are characterized by slow-cycling nature therefore less mitotic activities. To this end, we quantitatively evaluated Col2a1^CE^-tdT^+^ cells for EdU incorporation by flow cytometry. To this end, Col2-Q mice were fed with doxycycline for 1, 2 and 4 weeks and treated with tamoxifen twice at 3 and 2 days before analysis. These mice were pulsed with two doses of EdU shortly before analysis, at 6 and 3 hours prior to sacrifice. Actively proliferating cells that incorporate EdU were fluorescently marked by Click-iT Plus Alexa 647 Flow Cytometry kit. We quantified the percentage of EdU-Alexa647^+^ cells among LRCs (cells with top 10% H2B-EGFP brightness) and non-LRCs (cells with 10-50% H2B-EGFP brightness). Non-LRCs were significantly enriched for EdU^+^ cells, both at 1 week and 2 weeks of chase (EdU^+^ cells, 1 week: 2.04±1.67% of LRCs; 7.04±2.30% of non-LRCs, n=6 mice [p=0.002]; 2 weeks: 1.13±0.74% of LRCs; 2.80±1.00% of non-LRCs, n=8 mice [p=0.047]) (Figure 4d,e left and middle, Figure 4 —figure supplement S2a). We also found a trend that non-LRCs were enriched for EdU^+^ cells at 4 weeks of chase, although not statistically significant (EdU^+^ cells, 2.40±0.85% of LRCs; 4.34±1.89% of non-LRCs, n=4 mice [p=0.200] (Fg.4e, right, Figure 4 —figure supplement S2b)). Therefore, LRCs are relatively resistant to EdU incorporation, denoting their relatively quiescent status.”

4. Line 157-158: The authors stated that "Five rounds of digestion completely liberated cells from the growth plate, while cells on the articular surface were almost undigested" and this statement is supported by the image in Figure 1f. It is unclear why or if collagenase digestion cannot or did not liberate the articular chondrocytes. On close examination, the appearance (focus on the shape of the edge) of the articular cartilage also changed after digestion (even though judging from the SOC those two are probably not the same section). I suspect that some cells in the superficial zone of the articular surface might also have come off and if those cells would be included in downstream experiments like RNA-Seq. Although it looks from figure 1d that the articular cartilage is not labeled either red or green, but the superficial zone of the articular cartilage possesses cells that are quite similar to the resting zone (see for example Li, FASEB J. 2017) and, at least in theory, are those kinds of cells would retain the GFP-H2B. Interestingly, the most differentially expressed gene in RNA-Seq is Prg4, a well-known superficial zone marker but largely undetected in the resting zone (see Lui, 2015 Ped Res).

We agree with the reviewer that our collagenase digestion may dissociate articular chondrocytes. We therefore changed the clause “while cells on the articular surface were almost undigested” to “while cells on the articular surface were largely undigested” from the revised main text.

The important issue relevant to our downstream RNA-seq analyses is whether Col2^CE^-tdT^+^ cells that are harvested from the microdissected growth plate include articular chondrocytes, and if so, how much is the contamination. As we discussed to the Response #2 above, *Col2a1-creER* marked a far greater number of chondrocytes in the growth plate than those in the articular cartilage, both at P35 and P49, as shown in new Figure 3a,b.

Importantly, after five rounds of collagenase digestion, there was a significant reduction of Col2^CE^-tdT^+^ cells in the growth plate, but not in the articular cartilage (Figure 3a,b). From this quantitative data, we estimate that 99.1±1.4% of Col2^CE^-tdT^+^ cells that we observed in flow cytometry represent those derived from the growth plate. Thus, our dissociation protocol almost exclusively isolates Col2^CE^-tdT^+^ growth plate chondrocytes, essentially without any contamination of chondrocytes from the articular cartilage.

In light of this finding, it is intriguing that *Prg4* was identified as the most upregulated gene in LRCs, as the reviewer pointed out.

In the revised manuscript, we mentioned the morphological similarities between chondrocytes on the superficial zone of the articular cartilage and the resting zone of the growth plate, and that Prg4 is a well-known marker for chondrocytes of the superficial zone that is considered to largely undetected in the resting zone (Lui, 2015 Ped Res).

The following section has been added to the Results:

“Interestingly, the most highly upregulated gene in LRCs was Prg4, which is a marker for chondrocytes in the superficial layer of the articular surface (Lui et al., 2015). Chondrocytes on the superficial zone of the articular surface are morphologically similar with those at the resting zone of the growth plate (Li et al., 2017).”

Further, we assessed *Prg4* mRNA expression in the growth plate and the articular cartilage using RNAscope fluorescent *in situ* hybridization assays. We also evaluated *Col2a1* mRNA levels in both tissues as a positive control. At P21, we found that *Prg4* mRNA was abundant in the articular cartilage but was also present at a low level in the growth plate. Interestingly at 7 weeks of age, *Prg4* mRNA was present not only in the articular cartilage, but also in the growth plate, particularly in the resting zone. Therefore, our RNAscope analyses demonstrate that *Prg4* mRNA is present in the resting zone of the growth plate in early adulthood (new Figure 5c and new Figure 5 —figure supplement S4).

In conclusion, we believe that our flow cytometry-based approach to isolate chondrocytes from the growth plate is highly reliable, and that the results of our comparative RNA-seq analysis show the *bona fide* profile of chondrocytes in the resting zone (i.e. label-retaining chondrocytes, LRCs). Interestingly, our results show that *Prg4*, a canonical marker for the superficial zone of the articular cartilage, is also expressed in the resting zone in adulthood.

The following sentences have been added to the Results:

“First, we assessed Prg4 mRNA expressions in the growth plate and the articular cartilage using RNAscope fluorescent in situ hybridization analysis. At P21, Prg4 mRNA was indeed in resting zone chondrocytes, at a level far fainter than that of articular chondrocytes (Figure 5 —figure supplement S4a, arrows). By 7 weeks, Prg4 mRNA was also detected in the resting zone in a similar pattern (Figure 5c, arrows, Figure 5 —figure supplement S4b, arrows). Therefore, chondrocytes of the resting zone of the postnatal growth plate express Prg4 at a low level, supporting the validity of our comparative RNA-seq analysis.”

5. Line 169-171: the authors stated "Therefore, these findings demonstrate that LRCs can be effectively identified and isolated from postnatal Col2-Q growth plates by combined microdissection, enzymatic digestion and flow cytometry-based approaches". It appears what the authors are showing is that chasing by dox removal, over time, is effective in removing GFP signaling in the digested cell population. However, whether the LRCs can be effectively identified and isolated depends on how much GFP-H2B signal diminishes after 1 (or more) cell division and perhaps other factors. Please address this issue.

Thank you very much for raising this very important point. We agree with the reviewer that it is important to determine how much H2B-EGFP signal diminishes upon one cell division.

To address this issue, we provide a more quantitative measurement of the signal dilution of H2B-GFP following one round of cell division. We pulsed Col2-Q mice with two doses of EdU, at 3 and 6 hours prior to sacrifice, after two weeks of Doxy chase at P35. As shown in new Figure 1b, we calculated the corrected total cell fluorescence (CTCF) values of H2B-GFP signal of cells in the resting zone and their adjacent EdU^+^ cells in the proliferating zone. By doing so, we assume that an EdU-negative cell in the resting zone has undergone one cell division and become an EdU-positive cell in the proliferating zone.

We found that EdU^+^ cells in the proliferating zone have significantly decreased H2B-EGFP CTCF values compared to their preceding EdU-negative cells in the resting zone. The average fold change between EdU-negative and EdU-positive cells was 2.03, suggesting that H2B-EGFP intensity decreases by a factor of two following one cell division (new Figure 1b, c).

This result is completely as expected, as histone-bound EGFP should be distributed equally between the two daughter cells upon cell division.

The following sentences have been added to the Results:

“We labelled actively proliferating chondrocytes by pulsing these mice with EdU twice at 6 and 3 hours prior to sacrifice. By this approach, we expect to visualize an EdU-positive cell in the proliferating zone that is descended from an EdU-negative cell in the resting zone through one cell division (Figure 1b, left panel, arrowheads and bracket). The quantification of CTCF values through the entire column revealed that the cells in the proliferating zone (PZ) had weaker H2B-EGFP signal than those of the resting zone (RZ), and the cells in the pre-hypertrophic (PHZ) and hypertrophic zones (HZ) had yet weaker H2B-EGFP signal (Figure 1b, center panel, yellow numbers). The side-by-side comparison of CTCF values between EdU^+^ cells in the proliferating zone and their preceding EdU-negative cells in the resting zone further revealed that H2B-EGFP CTCF values decreased significantly upon one cell division (Figure 1c, P35: H2B-EGFP CTCF, EdU^-^ cells in RZ = 267,184.6±109,457.8; EdU^+^ cells in PZ = 131,892.0±83,344.8, n=4 mice, 24 cell-pairs [p<0.0001]). The average fold change between EdU-negative and EdU-positive cells was 2.03, suggesting that H2B-EGFP intensity decreases by a factor of two following one cell division (Figure 1b,c), as expected from the fact that histone-bound GFP is distributed equally between the two daughter cells upon cell division. Further, the cells with the top 10% brightness were localized to RZ at the top of the growth plate; while there were 27 cells in this given column of chondrocytes, the 3 brightest cells (374,012; 343,652; 264,697) were localized to RZ (Figure 1b, center panel, arrowheads). Thus, based on these quantitative histological data, we designated the cells with the top 10% brightness as LRCs.”

6. The efforts by the authors to identify and isolate the LRCs population and to perform RNA-Seq to assist with this effort were appreciated by the reviewers. The impact of technical factors (dissection, digestion, label, flow cytometry, etc) and the generalized handling of the cells could potentially affect the fidelity of gene expression study. This is not a situation unique to this study. How long after the growth plate was taken until the RNA were eventually isolated from the cells? What measures where taken to control for or account for these possible technical sources of artifactual signals?

Thank you very much again for raising this important point for discussion. We agree with the reviewers that our cells are subjected to multiple layers of technical manipulations before being profiled. However, we would like to point out that both LRCs and non-LRCs have been treated under the identical experimental manipulations, including dissection, digestion and FACS-sorting, up to the point that their transcriptomes are interrogated. Therefore, artifactual signals should be mitigated and minimized as long as we compare LRCs with non-LRCs within the same cell preparation.

Generally speaking, it takes approximately 6 hours to harvest RNAs after the growth plate is dissected. This includes 30 min. for microdissection, 150 min. for digestion, 45 min. for antibody staining, 60 min. for FACS-sorting and 60 min. for RNA isolation.

7. In their RNA-Seq experiment, the authors defined non-LRCs as the "GFP mid-low cells". What cells are these in terms of histological position in the growth plate? Are these mostly proliferative chondrocytes or includes the entirety of the PZ-PHZ-HZ cells? The expression profile seems to indicate these could a bit of everything (high in ColX, Ihh, Dlx5, etc). It would have been even more interesting if the non-LRCs are the direct progenies of the LRCs (like the one cell that just exited the resting zone).

Thank you very much for this great question. Based on our new quantitative data shown in new Figure 1b, non-LRCs appear to include a diverse array of cells within the column, including proliferating, pre-hypertrophic and hypertrophic chondrocytes, and perhaps some resting chondrocytes as well. As discussed in the preceding response, the cells with the top 10% brightness correspond to 3 cells (374,012; 343,652; 264,697) in the resting zone, in this given example. As the reviewer pointed out, the expression profile of non-LRCs is in line with their diverse composition of cells in PZ, PHZ and HZ.

One of the co-authors of this study (Dr. Takao Hirai) tried laser-capture microdissection (LCM) to compare the profile of LRCs and their direct progeny of the LRCs; however, he found that this was technically impossible to capture single-cells from a desired location of growth plate sections. It would be very interesting in future studies to apply emerging spatial transcriptomic approaches to profile these pairs of cells.

The following sentences have been added to the Results:

**“**While chondrocytes with the brightest H2B-EGFP signal were localized to the resting zone, their descendants showed increasingly diluted H2B-EGFP signals as they progressed toward the proliferating and pre-hypertrophic zones (Figure 1b, Figure 2e, right panel).”

8. In figure 3, the authors again utilized a nice combination of mouse models to test how canonical WNT signaling affects resting chondrocytes. Based on the results from figure 3d-g, it seems the effect is quite subtle, which is slightly disappointing. The conclusions from this find are described in an over zealous manner and should be tempered (e.g. "slightly impaired" in line 262).

We agree with the reviewers that the findings of our APC haploinsufficient study was somewhat disappointing. Therefore, we have updated the text in the following:

The following underlined word has been added to the Results:

“These data indicate that the formation and the expansion of PTHrP^+^ cells in the resting zone are slightly impaired when canonical Wnt signaling is activated in these cells.”

9. The fact that the authors identified genes like Rspo3, Sfrp5, and Dkk2, suggests WNT signaling but does not automatically mean canonical wnt/catenin signaling. There is a reasonable amount of literature connecting these WNT modulators to other non-canonical WNT, such as the PCP WNT pathway (e.g. Ohkawara, 2011 Dev Cell). In particular, PCP WNT pathway have been implicated in resting zone chondrocyte transition to the proliferative zone (e.g. Andrew Dudley's work in 2009 Development, Tracy Ballock's work in 2012 J Orth Res, and more recently Li, 2017 in eLife). It is understood that PCP WNT signaling is activated when the chondrocytes exit the resting zone and start forming columns, thus it is not surprising at all that a lot of WNT modulator/inhibitors are highly expressed in the resting zone. The statement in line 330 "no previous report ties WNT signaling to the maintenance of putative skeletal stem cell populations in the resting zone of the growth plate" is erroneous. While the authors have advanced the field (particularly testing how activating canonical wnt might impact resting zone in figure 3), it is not appropriate to ignore or downplay the prior literature in this area.

We would like to thank the reviewers for their insightful comments. We apologize for the oversight. We have summarized the role of non-canonical, PCP WNT signaling in facilitating resting-to-proliferating chondrocyte transition in the revised manuscript. We have also revised the Discussion in line with this.

The following section has been added to the Introduction:

“Non-canonical Wnt/planar cell polarity (PCP) signaling plays a key role in facilitating asymmetric divisions of resting chondrocytes. Oriented division, rearrangement and intercalation of chondrocyte clones in the resting zone, and their subsequent asymmetric divisions into their daughter cells aligned with the axis of growth are the hallmark characteristic of growth plate chondrocytes (Li et al., 2017). Non-canonical Wnt/PCP signaling is activated when the chondrocytes exit the resting zone and start forming columns, guiding the oriented cell division of resting chondrocytes into proliferating cells and their further expansion. In fact, misregulation of non-canonical Wnt/PCP signaling via dominant-negative forms of Frizzled receptors results in severe shortening of the growth plate (Hartmann and Tabin, 2000). Oriented cell division is sensitive to both high and low PCP activity mediated in part by Fzd7 (Li and Dudley, 2009, Li et al., 2017). In addition, Wnt5a signals to establish PCP in chondrocytes along the proximal-distal axis through regulation of Vangl2 (Gao et al., 2011, Randall et al., 2012). However, how resting chondrocytes are regulated by non-canonical Wnt/PCP signaling members, such as Rspo3 (Ohkawara et al., 2011) in addition to Dkk2 (Niehrs, 2006) and Fzd receptors, are unknown.”

The following underlines sentences have been added to the Discussion:

“In the resting zone, Apc haploinsufficiency led to increased β-catenin protein expression specifically in the resting zone including in PTHrP^+^ chondrocytes, and slightly decreased formation and expansion of PTHrP^+^ chondrocytes, reducing differentiation capabilities of these cells into columnar chondrocytes in the proliferating zone. It is important to note that the effect of β-catenin stabilization in PTHrP^+^ chondrocytes appears to be modest. Lines of studies demonstrate that non-canonical Wnt/planar cell polarity (PCP) signaling plays a key role in facilitating asymmetric divisions of resting chondrocytes. Therefore, our findings support the notion that canonical and non-canonical Wnt signaling pathways concertedly modulate PTHrP^+^ chondrocytes in the resting zone and regulate their differentiation.”

10. Figure 3d and lines 245 to 248: The image is unlikely enough to support their claim that b-catenin protein is upregulated in PTHrP-CE; APC-het. The authors are encouraged to show quantitative data for the claim.

Thank you very much for this important comment. We have now quantified b-catenin (CTNNB) protein levels in the resting zone on histological sections. We found that in CTNNB protein levels were significantly upregulated in the resting zone of PTHrP^CE^; APC^Het^ growth plates compared to that of Control growth plate. This data have been added as a new Figure 6e.

The following sentences have been added to the Results:

“Immunohistochemical analysis revealed that the β-catenin protein was significantly upregulated in the resting zone of APC cHet growth plates (CTNNB-Alexa 647 signal: Control, 62.1±33.3; APC cHet: 283.9±65.8, n=4 mice [p=0.03]), as well as in PTHrP^CE^tdTomato^+^ cells therein (Figure 6d, leftmost panels, arrows, and Figure 6e), indicating that Apc haploinsufficiency indeed slows β-catenin degradation and activated canonical Wnt signaling specifically in the resting zone of the growth plate.”